# Genome-wide meta-analysis of myasthenia gravis uncovers new loci and provides insights into polygenic prediction

Alice Braun [1,2], Sudhanshu Shekhar [3,4], Daniel F. Levey [5,6], Peter Straub[7,8], Julia Kraft [1,2], Georgia M. Panagiotaropoulou[1,2], Karl Heilbron[1,2], Swapnil Awasthi[1,2], Rafael Meleka Hanna[9,10], Sarah Hoffmann[9,10], Maike Stein [9,10,11], Sophie Lehnerer[9,10], Philipp Mergenthaler [9,10,12,13], Abdelrahman G. Elnahas[14], Apostolia Topaloudi[4], Maria Koromina [15,16,17], Teemu Palviainen [18], Bergrun Asbjornsdottir [19], Hreinn Stefansson[19], Astros Th. Skuladóttir [19], Ingileif Jónsdóttir [19], Kari Stefansson [19], Kadri Reis[14], Tõnu Esko[14], Aarno Palotie [18], Frank Leypoldt [20], Murray B. Stein [21,22], Pierre Fontanillas [23], Estonian Biobank Research Team*, 23andMe Research Team*, Jaakko Kaprio [18], Joel Gelernter [5,6], Lea K. Davis [7,8,17], Peristera Paschou[4], Martijn R. Tannemaat [24], Jan J.G.M. Verschuuren [24], Gregor Kuhlenbäumer[20,25], Peter K. Gregersen [26], Maartje G. Huijbers [24,27], Frauke Stascheit [9,10,29], Andreas Meisel [9,10,12,29] & Stephan Ripke [1,2,28,29] ✉

Myasthenia gravis (MG) is a rare autoantibody-mediated disease affecting the neuromuscular junction. We performed a genome-wide association study of 5708 MG cases and 432,028 controls of European ancestry and a replication study in 3989 cases and 226,643 controls provided by 23andMe Inc. We identified 12 independent genome-wide significant hits ($P < 5e^{-8}$) across 11 loci. Subgroup analyses revealed two of these were associated with early-onset (at age <50) and four with late-onset MG (at age ≥ 50). Imputation of human leukocyte antigen alleles revealed inverse effect sizes for late- and early-onset, suggesting a potential modulatory influence on the time of disease manifestation. We assessed the performance of polygenic risk scores for MG, which significantly predicted disease status in an independent target cohort, explaining 4.21% of the phenotypic variation ($P = 5.12e^{-9}$). With this work, we aim to enhance our understanding of the genetic architecture of MG.

Myasthenia gravis (MG) is an autoimmune disease of the neuromuscular junction with a prevalence of approximately 20 per 100,000 individuals[1,2]. Its core characteristics include muscle weakness and fatigue after physical activity, with some patients experiencing respiratory failure during acute symptom exacerbation[2]. Autoantibodies directed against the nicotinic acetylcholine receptor (AChR-Ab), found in 80% of patients, along with muscle-specific tyrosine kinase (MuSK-Ab) and low-density lipoprotein receptor-related protein 4 (LRP4-Ab) serve as diagnostic markers for MG. However, around 10% of patients do not show evidence of any known MG antibody[3,4]. MG is frequently categorized into early-onset (EOMG; <50 years old at disease manifestation) and late-onset (LOMG; ≥50 years

A full list of affiliations appears at the end of the paper. *Lists of authors and their affiliations appear at the end of the paper.
✉e-mail: sripke@broadinstitute.org

**Table 1 | Discovery GWAS, replication and combined meta-analysis results for MG**

| Nearest gene | Index SNP | CHR | BP (GRCh37) | A1/A2 | FRQ case | FRQ control | Proxy | Discovery | | | Replication | | | Discovery and replication | | |
|---|---|---|---|---|---|---|---|---|---|---|---|---|---|---|---|---|
| | | | | | | | | P | OR | SE | P-rep | OR-rep | SE-rep | P-comb | OR-comb | SE-comb |
| PTPN22 | rs2476601[b] | 1 | 114377568 | A/G | 0.121 | 0.13 | same | 2.77e-31 | 1.45965 | 0.0325 | 3.92e-14 | 1.311637 | 0.035159 | 3.25e-43 | 1.38944 | 0.0239 |
| MAGI3 | rs7522138[d] | 1 | 114217705 | A/G | 0.441 | 0.47 | same | 6.77e-09 | 0.88462 | 0.0212 | 2.91e-02 | 0.957836 | 0.022749 | 3.36e-08 | 0.91796 | 0.0155 |
| CHRNA1 | rs643350l[a,b] | 2 | 175616667 | G/A | 0.151 | 0.112 | same | 3.54e-07 | 1.17081 | 0.031 | 1.01e-05 | 1.14714 | 0.03172 | 2.69e-11 | 1.15917 | 0.0222 |
| CTLA4 | rs231779[b] | 2 | 204734487 | T/C | 0.386 | 0.451 | rs231770 | 1.24e-06 | 1.11149 | 0.0218 | 2.37e-05 | 1.09845 | 0.024511 | 6.87e-10 | 1.10572 | 0.0163 |
| TNIP1 | rs6681227[a,c,d] | 5 | 150447128 | G/T | 0.165 | 0.128 | same | 7.44e-08 | 1.16754 | 0.0288 | 2.07e-02 | 1.075955 | 0.031392 | 3.03e-08 | 1.12479 | 0.0212 |
| MHC | rs1264706[b] | 6 | 30063652 | C/G | 0.129 | 0.072 | same | 1.26e-34 | 1.61996 | 0.0393 | 1.92e-02 | 1.09196 | 0.041711 | 3.44e-25 | 1.34528 | 0.0286 |
| TBX18 | rs215918 | 6 | 85513783 | A/G | 0.405 | 0.374 | same | 3.42e-08 | 0.88754 | 0.0216 | 3.30e-03 | 0.938976 | 0.02322 | 3.84e-09 | 0.91101 | 0.0158 |
| RNASET2 | rs2301436[a,d] | 6 | 167437988 | T/C | 0.493 | 0.477 | same | 1.09e-07 | 1.11818 | 0.021 | 6.69e-03 | 1.05781 | 0.022716 | 2.33e-08 | 1.08992 | 0.0154 |
| FAM76B | rs4409785[b] | 11 | 95311422 | C/T | 0.192 | 0.169 | same | 2.73e-11 | 1.19494 | 0.0267 | 3.44e-03 | 1.08206 | 0.028953 | 1.48e-11 | 1.14168 | 0.0196 |
| ATXN2 | rs4766578[a] | 12 | 111904371 | T/A | 0.516 | 0.406 | same | 6.72e-08 | 1.12345 | 0.0216 | 2.09e-04 | 1.08362 | 0.022773 | 2.35e-10 | 1.1044 | 0.0157 |
| IKZF3 | rs12946510 | 17 | 37912377 | T/C | 0.476 | 0.493 | same | 7.61e-11 | 1.1474 | 0.0211 | 2.38e-04 | 1.08288 | 0.022772 | 8.28e-13 | 1.11717 | 0.0155 |
| TNFRSF11A | rs723926l[b] | 18 | 60005046 | A/C | 0.457 | 0.436 | same | 3.98e-20 | 1.21458 | 0.0212 | 4.43e-04 | 1.079214 | 0.022905 | 2.46e-19 | 1.15016 | 0.0156 |

The table presents the results from the discovery-, replication-, and combined inverse-variance-weighted fixed-effects meta-analyses.
BP (GRCh37) Base Pair position on Genome Reference Consortium Human Build 37, CHR chromosome, comb combined meta-analysis of discovery and control GWAS results, FRQ frequency.
[a]Reached genome-wide significance in leave-one-out discovery GWAS.
[b]Previously published as genome-wide significant locus.
[c]Only previously reported in EOMG.
[d]Would not withstand a multiple testing correction for 13 million tests ($P < 3.85e^{-9}$), accounting for the leave-one-out GWAS.

old at disease manifestation) due to variations in symptom spectrums such as a higher prevalence of thymic hyperplasia in EOMG[5,6].

To date, no large-scale twin studies of MG have been conducted, but previous genome-wide association studies (GWAS) have estimated the single-nucleotide polymorphism (SNP) based heritability to be around 25.6%[7]. While approximately 5% of patients have a family history of MG, 28.4% of cases report having relatives diagnosed with other autoimmune diseases[8–10].

Previous GWAS of MG have identified genetic loci at the MHC and close to *PTPN22* (Chr1:114377568), *CHRNA1* (Chr2:175629220), *CTLA4* (Chr2:204729153), *SFMBT2* (Chr10:7452743), *FAM76B* (Chr11:95311422), and *TNFRSF11A* (Chr18:60009814)[7,11–14]. Variants specific to EOMG were identified near *TNIP1* (Chr5:150440097) and *LINC02151* (Chr11:116028750)[11,14] while variants at *ZBTB10* (Chr8:81364205) were exclusively associated with LOMG[13]. MG has been linked to multiple human leukocyte antigen (HLA) alleles that play a crucial role in immune function[6,13–18], as well as to elevated complement system activation, particularly in AChR-Ab positive patients[19].

Understanding the genetic architecture of MG can provide crucial insights into pathogenesis which may aid in identifying individuals at risk and facilitate drug development efforts. Here, we present a comprehensive genetic analysis of MG, which includes a GWAS meta-analysis utilizing data from 12 cohorts and a replication sample provided by 23andMe Inc.

## Results

### Genome-wide association study of MG

We performed a GWAS meta-analysis of 5708 MG cases and 432,028 controls of European ancestry (effective $n_{half}$ = 9867), and 7,142,359 SNPs. Individual-level genotype data was available for 1927 cases. This discovery GWAS had a $\lambda_{GC}$ of 1.07 and an LDSC intercept of 1.017, indicating that a polygenic signal was the main source of inflation. The GWAS identified nine genome-wide significant index SNPs across eight loci and 24 index SNPs below a P-value of $1e^{-6}$ (Supplementary Figs. 1–3). Seven additional loci reached genome-wide significance across leave-one-out GWAS. Summary statistics for index SNPs with $P < 1e^{-4}$ are included in Supplementary Data 1.

We conducted a replication study of the index SNPs detected across discovery GWAS, leave-one-out GWAS, and previous publications in a 23andMe MG cohort of 3989 cases and 226,643 controls. Overall, we identified 12 independent genome-wide significant SNPs ($P < 5e^{-8}$) through the combined meta-analysis of discovery and replication GWAS, including six novel associations (Table 1, Fig. 1a). Furthermore, all seven index SNPs between $P < 1e^{-6}$ and $P > 5e^{-8}$ in the discovery and replication meta-analysis had the same direction of effect. A sign test indicated statistically significant enrichment of consistent effects ($P = 7.81e^{-3}$).

### Antibody-stratified association analysis

The meta-analysis includes cohorts restricted to AChR-Ab-positive MG cases ($n = 2798$) and ICD-based cohorts potentially incorporating patients from other antibody-mediated MG or seronegative cases ($n = 2910$). Notably, for the 12 reported index SNPs we found no significant differences in effect sizes between these two data sources except for the MHC locus (Supplementary Fig. 4).

### Genome-wide association study of early-onset MG

We additionally conducted a GWAS on a subset of 1391 EOMG cases utilizing the available phenotypic information. The control group included 22,407 individuals (effective $n_{half}$ = 2056). Our discovery GWAS of EOMG was performed on 7,542,347 SNPs ($\lambda_{GC}$ = 1.024) (Supplementary Fig. 5). We found genome-wide significant associations with four loci and seven additional index SNPs with $P < 1e^{-6}$. Summary statistics for SNPs with $P < 1e^{-4}$ are included in Supplementary Data 2.

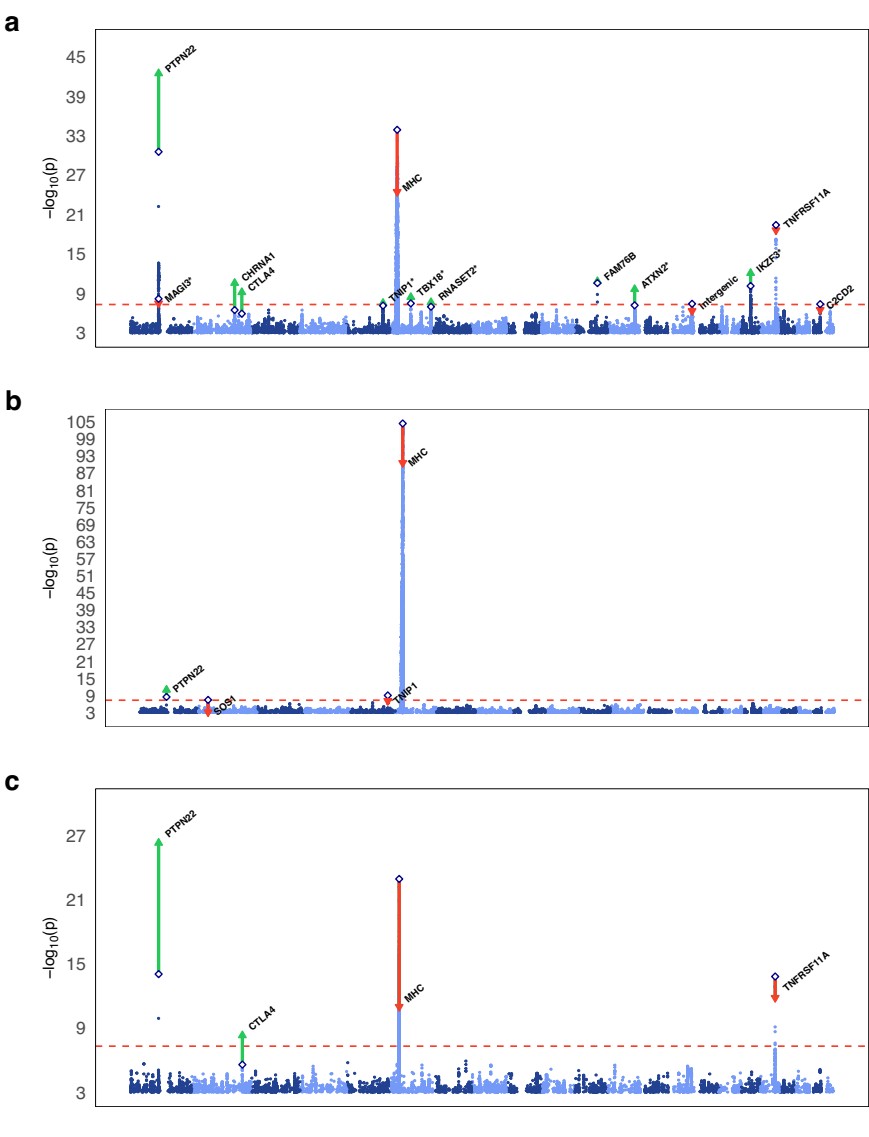

**Fig. 1 | Manhattan plots depicting GWAS results.** Plot **a** depicts the GWAS results for the full MG dataset, plot **b** shows the results of the early-onset MG GWAS, and plot **c** displays the results of the late-onset MG GWAS. The two-sided -log$_{10}$ P-values from the inverse-variance-weighted fixed-effects meta-analyses are plotted on the y-axis and the chromosomal position on the x-axis in ascending order from chromosome 1 through 22. The dashed red horizontal line marks the Bonferroni corrected genome-wide significance threshold ($P < 5e^{-8}$). Diamonds represent the index SNPs of the discovery GWAS. Red downward triangles indicate the index SNPs with a higher P-value in the replication and discovery meta-analysis while green upward triangles represent index SNPs with a lower P-value. Asterisks (*) indicate associations not previously reported.

The results were replicated in 871 23andMe cases that reported a diagnosis of EOMG and 109,843 control subjects, all of whom were under 50 years old. Overall, we report two loci that reached genome-wide significance in the meta-analysis of discovery and replication GWAS (Table 2). *SOS1*, which is the only unique locus not found in the combined MG GWAS, and *TNIP1* were not replicated (Table 1B, Fig. 1b, Supplementary Fig. 6).

### Genome-wide association study of late-onset MG

We further conducted a GWAS of 2404 LOMG cases and 64,103 controls not included in the EOMG GWAS (effective $n_{half}$ = 3701). The LOMG discovery GWAS included 6,310,403 SNPs ($\lambda_{GC}$ = 1.015) and identified three genome-wide significant loci (Supplementary Fig. 7) and four additional index SNPs below a P-value of 1e$^{-6}$. Results for all SNPs with $P < 1e^{-4}$ are included in Supplementary Data 3. A replication study was conducted in the 23andMe dataset of 3989 cases and 226,643 controls

because information on the age at diagnosis was unavailable. We found four genome-wide significant loci in the meta-analysis of discovery and replication, all of which overlapped with the combined MG GWAS meta-analysis (Table 3, Fig. 1c, Supplementary Fig. 8).

### Human leukocyte antigen results

We performed association analyses of 135 imputed HLA class I and II alleles in a subsample of 1927 cases and 5549 controls with available genotypes. Subsequently, we conducted separate analyses for EOMG (1080 cases and 3321 controls) and LOMG (846 cases and 2179 controls). Effects sizes for the strongest associations across the three subgroups are visualized in Fig. 2.

Our analysis revealed *HLA-B*08:01* as the top risk-conferring HLA allele for MG (OR = 2.349, $P = 1.15e^{-52}$, SE = 0.056; Supplementary Data 4, Supplementary Fig. 9). The top protective allele was *HLA-C*05:01*, (OR = 0.599, $P = 3.21e^{-10}$, SE = 0.082). After conditioning on

**Table 2 | Discovery GWAS, replication, and combined meta-analysis results for EOMG**

| Nearest gene | Index SNP | CHR | BP (GRCh37) | A1/A2 | FRQ case | FRQ control | Proxy | P | OR | SE | P-rep | OR-rep | SE-rep | P-comb | OR-comb | SE-comb |
|---|---|---|---|---|---|---|---|---|---|---|---|---|---|---|---|---|
| PTPN22 | rs2476601[a] | 1 | 114377568 | A/G | 0.156 | 0.136 | same | $3.61e^{-06}$ | 1.46976 | 0.0653 | $2.77e^{-04}$ | 1.30479 | 0.074616 | $1.15e^{-11}$ | 1.39585 | 0.0491 |
| MHC | rs2853986[a] | 6 | 31338844 | T/C | 0.682 | 0.907 | same | $4.06e^{-105}$ | 0.2395 | 0.0656 | $1.28e^{-11}$ | 0.636760 | 0.064406 | $2.67e^{-91}$ | 0.39404 | 0.046 |

The table presents the results from the discovery-, replication-, and combined inverse-variance-weighted fixed-effects meta-analyses.
BP (GRCh37) Base Pair position on Genome Reference Consortium Human Build 37, CHR chromosome, comb combined meta-analysis of discovery and control GWAS results, FRQ frequency.
[a]Previously published as genome-wide significant locus.

**Table 3 | Discovery GWAS, replication and combined meta-analysis results for LOMG**

| Nearest gene | Index SNP | CHR | BP (GRCh37) | A1/A2 | FRQ case | FRQ control | Proxy | P | OR | SE | P-rep | OR-rep | SE-rep | P-comb | OR-comb | SE-comb |
|---|---|---|---|---|---|---|---|---|---|---|---|---|---|---|---|---|
| PTPN22 | rs2476601[a] | 1 | 114377568 | A/G | 0.141 | 0.132 | same | $9.26e^{-15}$ | 1.47669 | 0.0503 | $3.92e^{-14}$ | 1.311637 | 0.035159 | $5.11e^{-27}$ | 1.3637 | 0.0288 |
| CTLA4 | rs231779[a,b] | 2 | 204734487 | C/T | 0.572 | 0.515 | rs231770 | $2.66e^{-06}$ | 0.85172 | 0.0342 | $2.37e^{-05}$ | 0.910374 | 0.024511 | $4.99e^{-09}$ | 0.89003 | 0.0199 |
| MHC | rs72848204[a] | 6 | 32594073 | G/T | 0.749 | 0.697 | same | $1.20e^{-23}$ | 1.50953 | 0.0411 | $2.37e^{-02}$ | 1.049218 | 0.024296 | $1.04e^{-11}$ | 1.15281 | 0.0209 |
| TNFRSF11A | rs7239261[a] | 18 | 60005046 | A/C | 0.492 | 0.368 | same | $1.56e^{-14}$ | 1.29576 | 0.0337 | $4.43e^{-04}$ | 1.079214 | 0.022905 | $1.50e^{-12}$ | 1.14339 | 0.0189 |

The table presents the results from the discovery-, replication-, and combined inverse-variance-weighted fixed-effects meta-analyses.
BP (GRCh37) Base Pair position on Genome Reference Consortium Human Build 37, CHR chromosome, comb combined meta-analysis of discovery and control GWAS results, FRQ frequency.
[a]Previously published as genome-wide significant locus.
[b]Would not withstand a multiple testing correction for 13 million tests ($P < 3.85e^{-9}$), accounting for the leave-one-out GWAS.

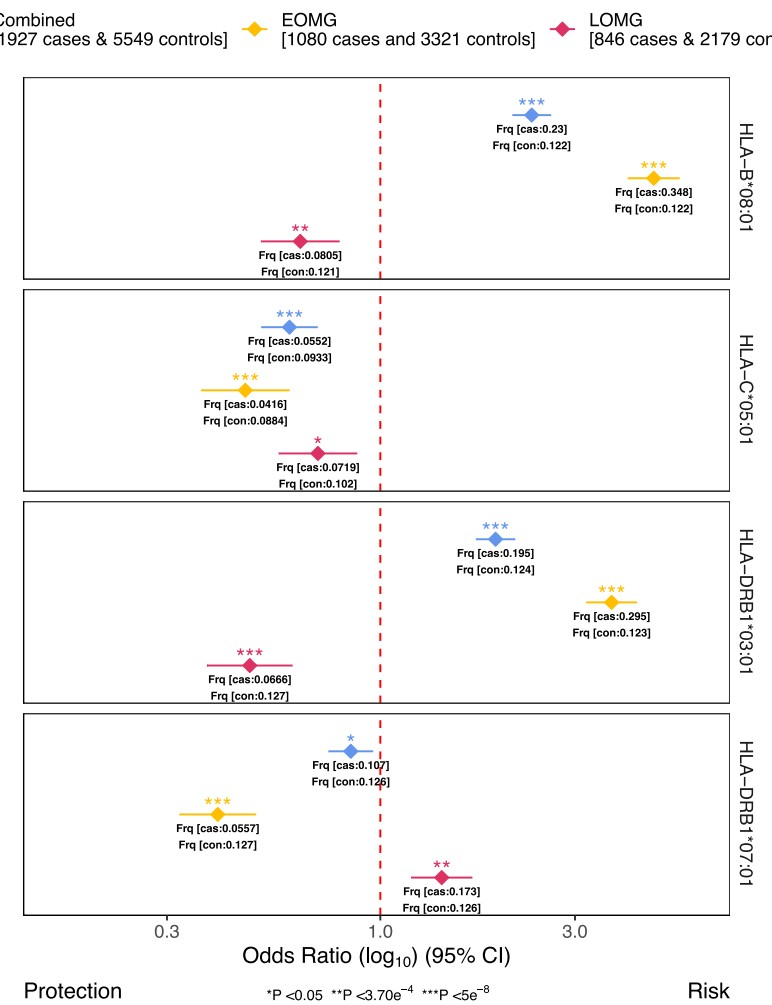

**Fig. 2 | HLA association analysis results.** The forest plot displays the top risk-conferring and protective HLA allele across all main analyses. The $\log_{10}$ of the odds ratio (OR) from inverse-variance-weighted fixed-effects meta-analysis is indicated as diamonds for each HLA allele and dataset used along with the 95% confidence intervals (error bars). Different $P$-value significance levels are indicated by asterisks for nominally significant ($P < 0.05^*$), Bonferroni-corrected ($P < 3.70e^{-4}$**), and genome-wide ($P < 5e^{-8}$***). MG myasthenia gravis; EOMG early-onset myasthenia gravis, LOMG late-onset myasthenia gravis.

the top variant rs4143332 (LD-$r^2$ of 0.988 to *HLA-B\*08:01*), or complement component 4 (*C4*) expression levels *HLA-C\*05:01* remained nominally significant (rs4143332: OR = 0.6869, $P = 5.662e^{-6}$, SE = 0.0828; *C4* expression: OR = 0.68991SE = 0.0865, $P = 1.774e^{-5}$), indicating partial independence from *HLA-B\*08:01* and *C4*. After conditioning on *C4A* and *C6B* expression levels *HLA-B\*08:01* remained genome-wide significant (OR = 2.34527, SE = 0.0777, $P = 5.69e^{-28}$).

*HLA-B\*08:01* was also the top risk-conferring allele associated with EOMG (OR = 4.677, P = $2.18e^{-94}$, SE = 0.075; Supplementary Data 5, Supplementary Fig. 10) while *DRB1\*07:01* was the top protective (OR = 0.3996, $P = 1.11e^{-16}$, SE = 0.111). After conditioning on the top variant rs2596565 (LD-$r^2$ of 1 to *HLA-B\*08:01*), the protective effect of *HLA-DRB1\*07:01* remained genome-wide significant (OR = 0.529, $P = 2.63e^{-8}$, SE = 0.115), indicating *HLA-DRB1* as an independent HLA-haplotype. When additionally conditioning on *C4* expression both effects remained genome-wide significant (*HLA-B\*08:01*: OR = 5.00531 SE = 0.0.1082 $P = 4.453e^{-50}$; *DRB1\*07:01*: OR = 0.34311, SE = 0.1334, $P = 1.048e^{-15}$)

The top HLA allele associated with LOMG was *HLA-DRB1\*03:01* which implied a protective effect (OR = 0.479, $P = 2.37e^{-9}$, SE = 0.123; Supplementary Data 6, Supplementary Fig. 11). After conditioning on *C4* expression the effect remained genome-wide significant (OR = 0.36400, SE = 0.1583 $P = 1.724e^{-10}$). We observed a significant opposite direction effect for *HLA-DRB1\*03:01* in MG (OR = 1.916, $P = 2.72e^{-30}$,

SE = 0.057) and much more pronounced in EOMG (OR = 3.689, $P = 6.12e^{-71}$, SE = 0.073). The top risk-conferring allele *DRB1\*07:01* in LOMG did not reach genome-wide significance (OR = 1.414, $P = 8.86e^{-5}$, SE = 0.088).

## Complement component 4 results
We performed association tests and meta-analyses of imputed *C4* haplotypes in 1927 MG cases and 5549 controls, 1080 EOMG cases and 3321 controls, and 846 LOMG cases and 2179 controls. Two isoforms of C4 are encoded by the genes *C4A* and *C4B* located at the MHC class III region. These vary in size and copy number combinations, resulting in long forms (L; -21 kilobases) and short forms (S; -14 kilobases). Specifically, we examined four common structural haplogroups of *C4A* (A) and *C4B* (B) (BS, AL-BS, AL-BL, and AL-AL). We calculated multiple logistic regression models with BS as the reference haplogroup, which has the fewest gene copy numbers. The resulting odds ratios ranged from 0.46–0.54 in MG, 0.21–0.27 in EOMG, and 1.83–2.27 in LOMG (Supplementary Data 7).

## Gene prioritization results
To link loci implicated by the GWAS to protein-coding genes, we applied positional mapping, expression quantitative trait loci (eQTL), and chromatin interaction gene mapping. This resulted in 52 mapped genes across 11 loci, excluding the extended MHC region. Four of these

loci (rs6433501, rs231779, rs6861227, rs215918) only contained a single protein-coding gene (*CHRNA1*, *CTLA4*, *TNIP1*, *TBX18*). Furthermore, our analyses highlighted the index SNP rs2476601 in *PTPN22* on chromosome 1 (PIP = 66%) and rs231775 in *CTLA4* on chromosome 2 (PIP = 14%; max. PIP = 19%; LD-$r^2$ = 0.97 with lead variant rs231779) as non-synonymous variants. On chromosome 11 there were no protein-coding genes within the defined locus boundaries. For the remaining loci, positional, eQTL, and chromatin interaction mapping nominated multiple genes in the loci, none of which can be prioritized with high confidence. Gene-level results are shown in Supplementary Data 8, and annotated results for SNPs in the credible set with PIP > 0.01 are included in Supplementary Data 9.

We performed drug target enrichment analysis on the set of 52 prioritized genes as input to the Genome for REPositioning drugs (GREP) pipeline[20]. Significant enrichment was observed only for the ATC category "ectoparasiticides, incl. scabicides, insecticides and repellents" (OR = 28.73 and Fisher's exact $P$ = 4.3e$^{-2}$). However, this result does not withstand multiple testing corrections, and the target gene *ALDH2* is situated within the complex chromosome 12 locus.

### Transcriptome-wide association study results

TWAS using the individual tissue-based prediction matrix of gene expression identified 138 significant genes whose transcript expression was significantly associated with MG. Permutation and co-localization tests identified 56 and 45 gene hits below the significance threshold. Combining TWAS, permutation, and co-localization tests, we identified 24 significant unique genes with high confidence. The TWAS highlighted individual genes in the highly complex loci on chromosome 12 and 17, as well as genes in additional loci that did not reach genome-wide significance in the GWAS, including *PAPPA* and *EPS15L1* (Supplementary Data 10, Supplementary Figs. 12–14).

### Polygenic risk scoring results

After excluding the test sample, our training GWAS comprised 5318 cases and 431,304 controls. Conservatively adjusting for 50 tests resulted in a Bonferroni-corrected $P$-value of <0.001. In the test sample of 390 cases and 724 controls the polygenic risk score (PRS) for MG explained 4.21% of variation in disease status ($R^2_{observed}$) at a $P$-value thresholds (PT) of <0.001 ($P$ = 5.12e$^{-9}$, AUC = 0.607, $R^2_{liability}$ = 0.96%; Fig. 3, Supplementary Data 11). In a sample of 176 EOMG cases and 613 age-matched controls, the MG-PRS performed best at a PT < 0.1. It explained 2.6% of the variation between early-onset cases and controls ($P$ = 2.58e$^{-4}$, AUC = 0.579, $R^2_{liability}$ = 0.64%; Supplementary Fig. 15A,

Supplementary Data 12). In a sample of 213 LOMG cases and 62 matched controls, the MG-PRS explained 7.6% of the variation between cases and controls at PT < 0.001 ($P$ = 3.36e$^{-4}$, AUC = 0.643, $R^2_{liability}$ = 1.75%; Supplementary Figs. 15B, Supplementary Data 13). We further assessed the performance of the MG-PRS when stratifying cases by antibody profile. We identified 73 cases that tested negative for AChR-Ab, including MuSK- and LRP4-Ab-positive patients. These cases were merged with 343 randomly selected controls. The MG-PRS explained 2.9% of the phenotypic variation at a PT of <0.5. However, the $P$-value remained above the Bonferroni-corrected threshold ($P$ = 7.4e$^{-3}$, AUC = 0.556, $R^2_{liability}$ = 0.71%; Supplementary Fig. 16A, Supplementary Data 14). Finally, we calculated the MG-PRS in 371 AChR-Ab-positive cases and 333 independent controls which explained 4.8% of phenotypic variation at a PT < 0.0001 ($P$ = 1.18e$^{-6}$, AUC = 0.620, $R^2_{liability}$ = 1.01%; Supplementary Fig. 16B, Supplementary Data 15).

### Genetic correlation and variant lookup results

We assessed the genome-wide genetic correlation ($r_g$) of MG, EOMG, and LOMG with 15 autoimmune and neurological traits via Linkage Disequilibrium Score Regression[21] (LDSC). We identified five significant genetic correlations with MG. The strongest correlations were observed with type-1 diabetes ($r_g$ = 0.523, $P$ = 1.42e$^{-6}$, SE = 0.109) and rheumatoid arthritis ($r_g$ = 0.5082, $P$ = 1.11e$^{-6}$, SE = 0.104). For EOMG, the strongest correlation was with systemic scleroderma ($r_g$ = 0.686, $P$ = 5.18e$^{-6}$, SE = 0.151), and for LOMG with vitiligo ($r_g$ = 0.442, $P$ = 9.69e$^{-5}$, SE = 0.113). The full results are presented in Supplementary Fig. 17 and Supplementary Data 16-18.

Genetic correlation analyses performed with 835 medical end-points in FinnGen R8 and MG highlighted 17 traits that passed the Bonferroni adjusted $P$-value of 5.88e$^{-5}$ (correcting for overall 850 tests). We found the strongest correlations between MG and auto-immune hyperthyroidism ($r_g$ = 0.586, $P$ = 6.27e$^{-7}$, SE = 0.118), and seropositive rheumatoid arthritis, strict definition ($r_g$ = 0.50, $P$ = 1.09e$^{-5}$, SE = 0.114). Results for all traits below nominal significance ($P$ < 0.05) are presented in Supplementary Data 19.

We have further assessed the genetic correlation between datasets ascertained through clinical studies without antibody-based inclusion criteria (effective $n_{half}$ = 507), clinical studies limited to AChR-ab cases (effective $n_{half}$ = 5,067) and electronic healthcare records (EHR) (effective $n_{half}$ = 4293). We observe the highest correlation between AChR-ab filtered and clinical datasets ($r_g$ = 0.6831, SE = 0.2137, $P$ = 0.0014). However, correlations between EHR based

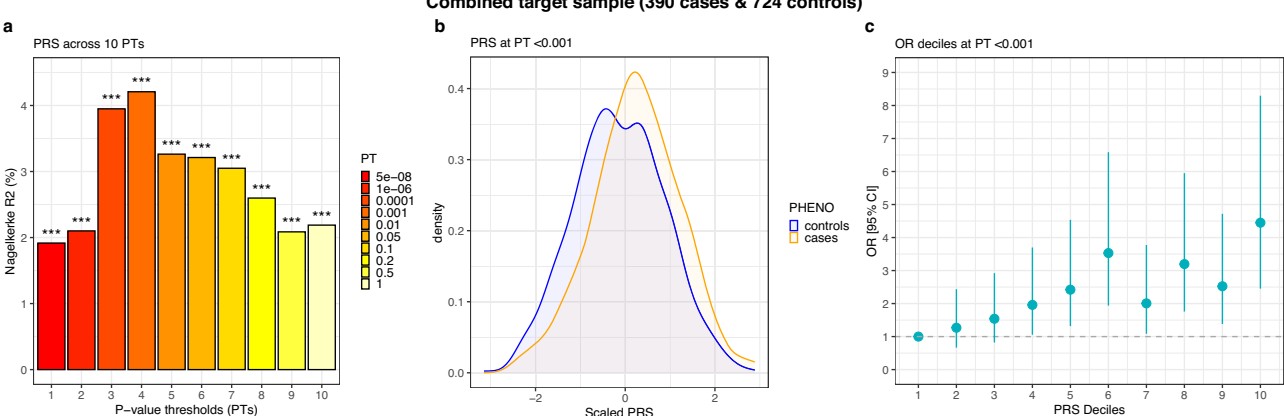

**Fig. 3 | Performance of MG polygenic risk scores.** Results of polygenic risk scoring of the combined target sample of CCM and LUMC cases and controls across all 10 PTs. Panel **a** shows the proportion of variance explained through the logistic regression models for PTs 1-10. Panel **b** shows the density distribution of the z-transformed best-performing PT ($P$ < 0.001) for cases (orange) and controls (blue). Panel **c** shows the odds ratios (OR) from logistic regression models for PT $P$ < 0.001 across ten deciles of the PRS along with the corresponding 95% confidence intervals (error bars). The target sample was scored using the combined MG leave-one-out training dataset of 5318 cases and 431,304 controls. PRS polygenic risk score.

and clinical samples ($r_g = 0.4774$, SE = 0.2430, $P = 0.0494$) and between EHR based and AChR-ab filtered ($r_g = 0.4021$, SE = 0.1323, $P = 0.0024$) were nominally significant.

We conducted a lookup of index SNPs included in the discovery GWAS and those in LD (LD-$r^2 > 0.1$) in GWAS Catalog to assess pleiotropy. Furthermore, GWAS Catalog, UK Biobank, and FinnGen R6 datasets were searched for associations with individual index SNPs. The lookups revealed large pleiotropy with other autoimmune diseases and immune-related molecular phenotypes, including white blood cell count. GWAS Catalog results for all SNPs with $P < 1e^{-4}$ are included in Supplementary Data 1. Genome-wide significant associations with other traits for each index SNP are shown in Supplementary Fig. 18.

### Heritability estimates

We have estimated the SNP-based heritability ($h^2_{SNP}$) of MG using LDSC[22] and the genome-wide complex trait analysis[23] (GCTA) software package. The $h^2_{SNP}$ on the liability scale based on the 7476 individuals with available genotypes measured through GCTA was 0.1818 (SE = 0.0149). The $h^2_{SNP}$ for the same 7476 individuals was 0.1184 (SE = 0.0272) using LDSC. The $h^2_{SNP}$ on the liability scale as measured by LDSC based on the summary statistics of the whole sample of 437,736 individuals was 0.0552 (SE = 0.0091).

## Discussion

The current GWAS represents the largest genetic study of MG to date, tripling the sample size of prior studies and encompassing all published genotyped samples of European ancestry[7,11–14]. Novel associations in overall MG included variants close to *MAGI3* (Chr1:114217705), *TBX18* (Chr6:85513783), *RNASET2* (CHR6:167437988), *ATXN2* (Chr12:111904371), *IKZF3* (Chr17:37912377), and finally *TNIP1* (Chr5:150447128), which has been implicated in EOMG before[14]. Moreover, we confirm six previously reported associations[7,11,12]. The MHC as well as variants close to *PTPN22*, *CTLA4*, *TNIP1*, *FAM76B*, *IKZF3*, and *TNFRSF11A* are well-established loci implicated in immune function and disease[24–29]. Among the novel loci, *IKZF3*, which encodes a transcription factor involved in regulating lymphocyte differentiation and activation, has been linked to rheumatoid arthritis[30,31] and systemic lupus erythematosus[32]. *ATXN2* was found to be involved in neurodegeneration[33] and thyroid disease[34]. However, some of these loci are complex, making the causal gene difficult to pinpoint. We employed multiple mapping techniques to associate GWAS-identified loci with protein-coding genes, identifying 52 mapped genes across 11 loci, excluding the MHC region, which we analyzed separately. Notably, four loci contained only a single gene. First, *CHRNA1*, a nicotinic acetylcholine receptor subunit and a target of neuromuscular blocking agents[35]. Second, *CTLA4*, a member of the CD28 immunoglobulin superfamily regulating T-cell responses and an immunotherapy target[36]. Third, *TNIP1*, which plays a key role in inflammatory processes showed a high loss-of-function intolerant probability (pLI) of 0.93. Fourth, *TBX18*, involved in embryonic development[37], which had a high pLI score of > 0.99 and mapped SNPs reached a maximum CADD score of 18.77, indicating deleteriousness[38,39]. Furthermore, our analyses highlighted rs2476601 in *PTPN22* on chromosome 1 and rs231775 in *CTLA4* on chromosome 2 as nonsynonymous variants.

The MHC represents one of the strongest signals in MG across subtypes. Our analyses identified *HLA-B\*08:01* as the top associated allele, replicating the results of smaller-scale studies[14,17,40–42]. While we confirm a protective effect of *HLA-DQB1\*03* and a risk-conferring effect of *HLA-DRB1\*07* on LOMG[15] we found opposite direction effects in EOMG. Inverse odds ratios based on the onset subtype were additionally observed for *HLA-B\*08:01* and *DRB1\*03:01*. Given that the vast majority of cases in this analysis tested positive for AChR-Ab (~97% of EOMG and ~96% of LOMG cases), it is unlikely that antibodies drive these effects. These findings could imply a modulating effect of HLA alleles on the time of disease onset. However, due to the complex LD

structure of the MHC, independent large-scale sequencing studies are needed to dissect the functionality of individual HLA alleles in the etiology of MG. The link between *C4* allele copy number and MG risk suggests a contribution of complement activity variation to the auto-immune response targeting the neuromuscular junction, particularly in AChR-Ab-positive MG patients, where complement activation is more prominent[19,43].

We demonstrated that the PRS for MG predicted disease status in an independent sample while performing better at *P*-value thresholds above genome-wide significance. This indicates that more loci are likely to be discovered with increasing sample size and statistical power.

We replicated the genetic correlation of MG with type 1 diabetes, rheumatoid arthritis, and vitiligo[12] and further found positive correlations of MG with systemic lupus erythematosus and multiple sclerosis. LOMG was most strongly associated with vitiligo and EOMG showed a strong association with systemic sclerosis. In a less restrictive analysis utilizing the FinnGen R8 dataset, autoimmune hyperthyroidism emerged as the strongest correlation. This analysis further confirmed the correlation with rheumatoid arthritis. However, these correlations could be driven by the presence of autoimmune comorbidity. Future studies could further investigate the results, e.g., by excluding co-morbid individuals. We detected significant genetic correlation among all data sources, EHR, clinical samples, and clinical samples limited to AChR-ab positive cases. As would be expected, the strongest correlation was found between the more rigorously collected clinical samples and AChR-ab positive cases, underscoring some variability introduced by the EHR collections.

We present $h^2_{SNP}$ estimates ranging from 0.056 to 0.18 on the liability scale, obtained using LDSC and GCTA GREML. Our GCTA estimate in clinically-ascertained samples ($n = 7,476$) was similar to the value originally reported by Renton and colleagues[7] ($h^2_{SNP} = 0.26$). Our LDSC estimate in the same sample was lower, which was expected given that LDSC $h^2_{SNP}$ estimates tend to be smaller than those from GCTA and capture the lower bound of $h^2_{SNP}$[44]. Our LDSC $h^2_{SNP}$ estimate (0.0552) was lower still when including samples from large biobanks. This may be due to a higher rate of misdiagnosis in biobank cases, and/or due to clinically-ascertained cases and controls representing a more severe phenotype[45].

Our analyses have multiple limitations to be considered. Firstly, antibody phenotypes were not available for all cohorts. A GWAS stratified by antibody profiles could reveal distinct etiological differences, considering the significant impact of antibody subtypes on therapeutic stratification, future studies should concentrate on the currently low number of genotyped AChR-Ab negative MG patients. The diagnoses in the replication sample were based on self-reporting, which could potentially render them less reliable compared to diagnoses obtained through clinical ascertainment. Furthermore, conditional analyses conducted by employing *C4* gene expression may not fully account for the LD with HLA alleles, which were imputed using a different reference panel. Sequencing studies are needed to confirm the observed effects and establish their independence. Finally, our analyses are based on European ancestry samples, limiting the discovery rate and the predictive capability of PRS across ancestries. In the next step, we aim to expand our analyses to include samples from non-European ancestries.

The presented GWAS identified 12 genome-wide significant index SNPs associated with MG, contributing to an advancement in our comprehension of MG genetics. Our study elucidated the role of HLA alleles on disease onset and demonstrated that the PRS for MG effectively predicts MG in an independent sample, suggesting its potential to complement current diagnostic tools.

## Methods

### Sample ascertainment

We obtained genotypes from two unpublished and three published GWAS[7,13,14]. We additionally received summary-level data from two

previous GWAS[11,12], and from the UK Biobank[46], the Million Veteran Program[47,48], deCODE genetics[49], the Estonian Biobank[50], FinnGen[51], and BioVU[52].

Information on age of onset was available for a subset of cohorts resulting in a smaller sample size of 1,391 cases and 22,407 controls in the early-onset GWAS and 2,404 cases along with 64,103 controls in the late-onset GWAS. We calculated half of the effective sample size ($N_{eff}$) for each cohort separately using the following formula: $(4 \times Ncases \times Ncontrols/(Ncases+Ncontrols))/2$ and summed it across all cohorts.

Antibody phenotypes were available for 3188 cases, of which 3115 were AChR-Ab positive. We had access to the serotype and genotype information of 73 confirmed non-AChR-Ab cases (MuSK-Ab- and LRP4-Ab positive as well as seronegative), rendering these samples too small for specific genetic association analyses. Information on previous publications, sample sizes and phenotypes per cohort are shown in Supplementary Data 20–22.

## Inclusion criteria

Cases met the international classification of diseases (ICD) diagnostic criteria for MG (ICD-10 code G70.0 or ICD-9 code 358.0). Newly obtained control samples were filtered for common autoimmune diseases due to the overall high genetic correlation with MG[27] (Supplementary Data 23). All individuals included in the analyses were of European ancestry and provided written informed consent. All included cohorts received approval by their respective local institutional review boards. More detailed sample descriptions are included in the Supplementary Information.

## Genomic quality control and imputation

Quality control and imputation were performed using software implemented in the Rapid Imputation for Consortias Pipeline (RICOPILI)[53] for all cohorts that provided genotypes. We applied standard quality filters to retain individuals and SNPs. These filters include SNP missingness <0.05, subject missingness <0.02, autosomal heterozygosity deviation $F_{het} < 0.2$, SNP missingness after sample exclusion <0.02, minor allele frequency (MAF) > 0.01, a difference in SNP missingness between cases and controls <0.02, and Hardy-Weinberg equilibrium of $P > 10e^{-6}$ in controls and $P > 10e^{-10}$ in cases. Related individuals were filtered based on identity by descent segments (PI-HAT > 0.2). One member of the related pairs was retained while cases were preferred to control subjects. Externally generated GWAS summary statistics were aligned to Genome Reference Consortium Human Build 37 and filtered to include only SNPs with a MAF > 0.01 if necessary. A principal component analysis of the genotyped SNPs was performed via EIGENSTRAT[54] and plotted to identify and exclude ancestral outliers by visual inspection. After quality control, all cohorts showed a genomic inflation factor ($\lambda_{GC}$) of 1.034–1.073. Imputation was carried out using the Haplotype Reference Consortium reference panel release 1.1[55]. Haplotype pre-phasing and imputation were performed using EAGLE version 2.4.1[56] and MINIMAC3[57].

## Genome-wide association meta-analyses

We ran GWAS on imputed dosage files of SNPs with a MAF > 0.01 using additive logistic regression models for MG versus controls, LOMG versus controls, and EOMG versus controls. By default, we included the first six principal components (PCs) in the models as covariates. Results were meta-analyzed using METAL[58] with the effect size estimates weighted by the inverse of the corresponding standard errors (SE). We defined an independent signal as the area around an index SNP with a P-value $< 5e^{-8}$ and a linkage disequilibrium (LD) of $r^2 < 0.1$ within a 3-megabase window. We conducted leave-one-out GWAS to identify additional significant candidate loci for replication that were potentially lost due to heterogeneity in our overall sample. All newly identified or previously reported GWAS loci were subsequently confirmed or rejected via replication analyses, including meta-analysis and binomial sign-tests.

## Replication sample

We replicated our results in a sample ascertained through 23andMe, Inc. The diagnosis of MG was self-reported by the participants. Control individuals were selected at random, constituting 5% of the 23andMe control group, to enhance computational efficiency while minimizing power reduction. We defined EOMG cases as those under 50 years old at participation. 871 individuals met this criterion and were merged with 109,843 randomly drawn controls younger than 50. As no age of onset information was available, we replicated the LOMG discovery GWAS loci using the full 23andMe MG dataset.

## Human leukocyte antigen imputation

For all available genotypes, we imputed HLA alleles via a European 1000 Genomes[59] phase 3 reference panel of 503 individuals with HLA types inferred from sequencing data[60]. The reference was downloaded from the CookHLA[61] GitHub repository and includes 151 HLA alleles with a frequency > 0.01. Pre-phasing and imputation were carried out via SHAPEIT2 and IMPUTE4[62]. We conducted association analyses with the same three dichotomous outcomes as in the GWAS on imputed dosage files. Conditional analyses were conducted via the stepwise inclusion of variants with the lowest P-value as covariates in logistic regression models until no signal below $P < 1e^{-6}$ was left. To further account for the complex LD structure of the MHC we conducted additional conditional analysis by calculating the $C4A$ and $C4B$ expression levels for each individual based on imputed $C4$ alleles. We have used the formula proposed by Sekar and colleagues[63]: $C4A$ expression = $(0.47 * AL) + (0.47 * AS) + (0.20 * BL)$; $C4B$ expression = $(1.03 * BL) + (0.88 * BS)$.

## Complement component 4 imputation

We used a reference panel based on 1265 sequenced individuals from the Genomic Psychiatry Cohort[64] to infer $C4$ haplogroups, which were imputed using Beagle 5.4[65]. We initially filtered the imputed $C4$ variants based on the gene composition levels (e.g., BS, AL-BS, AL-BL, AL-AL) by allele frequency (>0.01) in controls and imputation quality (INFO > 0.80). Subsequently, we removed individuals with dosage sums ≤1.9 to avoid these being attributed to the reference group in the logistic regression model. Dosages for all remaining $C4$ variants were then modeled together in a logistic regression for each outcome, incorporating the first six principal components and excluding the BS reference group, denoting the shortest $C4$ variant. Fixed-effects meta-analyses for MG, EOMG, and LOMG were conducted in R version 4.3.2[66] using meta version 7.0–0[67].

## Gene prioritization

In order to map SNPs within the identified loci to protein-coding genes we applied the SNP2GENE module implemented in FUMA version 1.5.2[68] via positional (10 kilobase window), eQTL (GTEx version 8 tissues, database of immune cell expression eQTLs), and chromatin interaction mapping (HiC of adult and fetal cortex, and GSE87112 tissues). We used the discovery GWAS summary statistics along with the set of 11 pre-defined index SNPs as input files for FUMA. Due to its complex LD structure we excluded the extended MHC region (Chromosome 6: 25–35 Mb) from all analyses. We additionally filtered the mapped genes with the LD-based locus boundaries defined by RICO-PILI (LD-$r^2 > 0.1$). Furthermore, we intersected FUMA functional annotations with SNPs in the 95% credible set with a PIP > 0.01 (calculated via the R package coloc 5.2.3 finemap.abf module[69]).

To evaluate the enrichment of clinical indication categories (ICD-10 and ATC) of druggable target genes in our GWAS, we utilized the set of genes prioritized by FUMA as input to the GREP pipeline[20]. The

pipeline employs Fisher's exact tests to determine whether the gene set demonstrates enrichment in genes targeted by medications.

### Transcriptome-wide association study

Transcriptome-wide association study (TWAS) was performed using FUSION[70] to predict tissue-specific gene expression based on the MG discovery GWAS summary statistics excluding the extended MHC region (Chromosome 6: 25–35 Mb). Gene expression weights from three Genotype-Tissue Expression (GTEx) version 8[71] tissues were used for transcriptomic imputation and association testing: Muscle Skeletal ($n = 8602$), Nerve Tibial ($n = 11,360$), and Whole Blood ($n = 8059$). Gene associations were considered significant at a $P$-value $< 0.05/N$ of genes per tissue. A permutation test, which shuffles the quantitative trait loci (QTL) weights ($N_{max} = 1000$) was performed to correct inflated associations from by-chance QTL co-localization. Co-localization was computed for genes with a P-value $< 1e$-5. A posterior probability value $\geq 0.75$ was considered as evidence for the expression-QTL-GWAS pair influencing both the expression and the GWAS trait in a particular region.

### Polygenic risk scoring

PRS analyses were performed to assess the capacity of the discovery results to predict MG, EOMG, and LOMG in an independent sample. Additionally, we evaluated PRS performance in AChR-Ab positive- and negative cases. We performed a leave-one-out GWAS meta-analysis to generate independent training and test datasets. PRS for MG were calculated via the LD-clumping and $P$-value thresholding method in PLINK 1.9[72]. The training dataset was clumped to account for LD, retaining only the most significant SNP within 500 kilobases and an LD of $r^2 > 0.25$. PRS was generated across ten $P$-value thresholds (PT): $5e^{-8}$, $1e^{-6}$, 0.0001, 0.001, 0.01, 0.05, 0.1, 0.2, 0.5, 1. The first six PCs were included in all logistic regression models as covariates. Control subjects were matched by age for onset-specific analyses and split randomly in antibody-specific analyses. The observed phenotypic variation was measured via Nagelkerke's pseudo-$R^2$ and the predictive performance via the area under the receiver operator curve (AUC).

### Genetic correlation analyses and variant lookup

The bivariate genetic correlations of MG, EOMG, and LOMG with 15 pre-selected neurological and autoimmune traits were computed via LDSC[73]. We downloaded publicly available summary statistics[26,31,74–85] from the GWAS catalog website [https://www.ebi.ac.uk/gwas/][86] and formatted summary statistics from Dr. Alkes Price's research group's repository [https://alkesgroup.broadinstitute.org/sumstats_formatted][22].

We have additionally performed genetic correlation analyses between MG and 835 medical endpoints, utilizing data from digital health record data of the 342,499 participants included in FinnGen Release 8[51]. We limited the medical endpoints to those with at least one genome-wide significant hit in FinnGen, indicating genetic susceptibility to the trait and ensuring an adequate sample size. We have excluded the FinnGen sample from our MG summary statistics for this purpose.

In order to assess pleiotropy, we conducted a lookup of all SNPs in the discovery GWAS and those in LD (LD-$r^2 > 0.1$) in GWAS Catalog (version from September 2018)[86] for associations with other traits. Additionally, we performed a lookup for the 11 individual index SNPs highlighted by the combined meta-analysis, excluding the MHC in data sources aggregated by Open Targets Genetics. The data sources include associations identified by the SAIGE study and the Neale lab conducted in the UK Biobank, summary statistics from GWAS Catalog, and FinnGen Release 6[87].

### Heritability estimates

We used LDSC[22] and the genome-based restricted maximum likelihood (single-component GREML) module implemented in GCTA[23] to assess $h^2_{SNP}$ of MG in our sample. We used summary statistics including all samples in the discovery GWAS and samples with available genotypes to estimate $h^2_{SNP}$ via LDSC. We further merged the PLINK files of all available genotypes as input for GCTA along with the first six PCs. We assumed a population prevalence of 20 per 100,000 individuals to transform the heritability estimates into the liability scale.

### Reporting summary

Further information on research design is available in the Nature Portfolio Reporting Summary linked to this article.

## Data availability

The summary statistics for the discovery GWAS meta-analyses of myasthenia gravis, early-onset myasthenia gravis and late-onset myasthenia gravis generated in this study have been deposited in GWAS catalog under accession codes GCST90432156, GCST90432157, GCST90432158. A subset of individual-level genetic data used in this article were downloaded through the database of Genotypes and Phenotypes (dbGaP). These include the Renton et al. [7] case dataset (accession code phs000726) [https://www.ncbi.nlm.nih.gov/projects/gap/cgi-bin/study.cgi?study_id=phs000726.v1.p1], the controls merged with the Renton et al. [7] cases (accession code phs000196): [https://www.ncbi.nlm.nih.gov/projects/gap/cgi-bin/study.cgi?study_id=phs000196.v3.p1], and the controls merged with the Gregersen et al. [14] cases (accession code phs000882). [https://www.ncbi.nlm.nih.gov/projects/gap/cgi-bin/study.cgi?study_id=phs000882.v1.p1]. Other individual-level data used in this study will be available upon request through a data transfer agreement with the respective institutions responsible for the data. Other publicly available data used in this study include the 1000 Genomes Project HLA reference [https://github.com/WansonChoi/CookHLA/tree/master/1000G_REF], the Chia et al.[11] summary statistics (GWAS Catalog accession code GCST90093061) [https://www.ebi.ac.uk/gwas/studies/GCST90093061], the FinnGen data release 8: [https://www.finngen.fi/en/access_results], LDSC formatted summary statistics [https://alkesgroup.broadinstitute.org/sumstats_formatted/] and the Haplotype Reference Consortium reference panel release 1.1: [https://ega-archive.org/studies/EGAS00001001710].

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

## Acknowledgements

We want to acknowledge the participants and investigators of the FinnGen study, the UK Biobank, the Million Veteran Program, BioVU, the Estonian Biobank, deCODE genetics, and the International Multiple Sclerosis Genetics Consortium for sharing data with us. We would like to thank the research participants and employees of 23andMe for making this work possible. Following biobanks are acknowledged for delivering biobank samples to FinnGen: Auria Biobank (www.auria.fi/biopankki), THL Biobank (www.thl.fi/biobank), Helsinki Biobank (www.helsinginbiopankki.fi), Biobank Borealis of Northern Finland (https://www.ppshp.fi/Tutkimus-ja-opetus/Biopankki/Pages/Biobank-Borealis-briefly-in-English.aspx), Finnish Clinical Biobank Tampere (www.tays.fi/en-US/Research_and_development/Finnish_Clinical_Biobank_Tampere), Biobank of Eastern Finland (www.ita-suomenbiopankki.fi/en), Central Finland Biobank (www.ksshp.fi/fi-FI/Potilaalle/Biopankki), Finnish Red Cross Blood Service Biobank (www.veripalvelu.fi/verenluovutus/biopankkitoiminta), Terveystalo Biobank (www.terveystalo.com/fi/Yritystietoa/Terveystalo-Biopankki/Biopankki/) and Arctic Biobank (https://www.oulu.fi/en/university/faculties-and-units/faculty-medicine/northern-finland-birth-cohorts-and-arctic-biobank). All Finnish Biobanks are members of BBMRI.fi infrastructure (www.bbmri.fi). Finnish Biobank Cooperative -FINBB (https://finbb.fi/) is the coordinator of BBMRI-ERIC operations in Finland. The Finnish biobank data can be accessed through the Fingenious® services (https://site.fingenious.fi/en/) managed by FINBB. The FinnGen project is funded by two grants from Business Finland (HUS 4685/31/2016 and UH 4386/31/2016) and the following industry partners: AbbVie Inc., AstraZeneca UK Ltd, Biogen MA Inc., Bristol Myers Squibb (and Celgene Corporation & Celgene International II Sàrl), Genentech Inc., Merck Sharp & Dohme LCC, Pfizer Inc., GlaxoSmithKline Intellectual Property Development Ltd., Sanofi US Services Inc., Maze Therapeutics Inc., Janssen Biotech Inc, Novartis AG, and Boehringer Ingelheim International GmbH. This research is based, in part, on data from the Million Veteran Program, Office of Research and Development, Veterans Health Administration. Funding for D.F.L. was provided by a Career Development Award CDA-2 from the Veterans Affairs Office of Research and Development (1IK2BX005058-01A2). Funding for Murray.S. and J.G. was provided from a Veterans Affairs Office of Research and Development Merit Award (I01CX001849). One dataset used for the analyses described were obtained from Vanderbilt University Medical Center's BioVU

which is supported by numerous sources: institutional funding, private agencies, and federal grants. These include the NIH funded Shared Instrumentation Grant S10RR025141; and CTSA grants UL1TR002243, UL1TR000445, and UL1RR024975. Genomic data are also supported by investigator-led projects that include U01HG004798, R01NS032830, RC2GM092618, P50GM115305, U01HG006378, U19HL065962, R01HD074711; and additional funding sources listed at https://victr.vumc.org/biovu-funding/. P.M. is Einstein Junior Fellow funded by the Einstein Foundation Berlin and acknowledges funding support by the Einstein Foundation Berlin (EJF-2020–602; EVF-2021–619, EVF-BUA-2022-694) and the Leducq Foundation for Cardiovascular and Neurovascular Research (Consortium International pour la Recherche Circadienne sur l'AVC). M.G.H. receives financial support from the LUMC (Gisela Thier Fellowship 2021), Top Sector Life Sciences & Health to Samenwerkende Gezondheidsfondsen (LSHM19130), Prinses Beatrix Spierfonds (W.OR-19.13). The LUMC is part of the European Reference Network for Rare Neuromuscular Diseases [ERN EURO-NMD] and the Netherlands Neuromuscular Center. S.R. has received funding from the German Research Foundation (Deutsche Forschungsgemeinschaft - DFG) (grant number 461427996). The Estonian Biobank work was supported by Personal research funding: Team grant PRG1291. S.R. and A.M. received funding from the German Research Foundation (Deutsche Forschungsgemeinschaft) via the Clinical Research Group KFO 5023 BeCAUSE-Y.

## Author contributions

A.M., A.B., F.S., and S.R. conceived and planned the study. A.B., S.S., D.F.L., and P.S. conducted statistical analyses. A.B. and S.R. meta-analyzed and interpreted the data. J.Kraft., G.M.P, M.K., and S.A. created software pipelines used in the work. G.K., F.L., and J.Kraft contributed to data processing. F.S., A.M., R.M.H., S.H., Maike.S., S.L., P.M., M.R.T., J.J.G.M.V., M.G.H. were involved in sample acquisition and data collection. P.K.G., A.G.E, Murray.S., J.G., A.T., T.P., P.P., B.Á., H.S., A.T.S., I.J., K.S., K.R., T.E., L.K.D., A.P., J.Kaprio, K.H., P.F., 23andMe Research Team, Estonian Biobank Research Team provided summary-level data. S.R. supervised the statistical analysis. A.B. drafted the manuscript. All authors carefully reviewed the manuscript and have approved the submitted version.

## Funding

## Competing interests

The authors declare the following competing interests: A.M. has received speaker or consultancy honoraria or financial research support (paid to his institution) from Alexion Pharmaceuticals, argenx, Axunio, Destin, Grifols, Hormosan Pharma, Janssen, Merck, Octapharma, UCB, and Xcenda. He serves as medical advisory board chairman of the German Myasthenia Gravis Society. A.Th.S., B.Á, H.S., I.J., and K.S. are employees of deCODE/Amgen Inc. F.S. has received speaker's honoraria from argx and Alexion, as well as honoraria for attending advisory boards for Alexion and UCB Pharma. F.L. is supported by the German Ministry of Education and Research (01GM1908A und 01GM2208), E-Rare Joint Transnational research support (ERA-Net, LE3064/2-1), Stiftung Pathobiochemie of the German Society for Laboratory Medicine and HORIZON MSCA 2022 Doctoral Network 101119457 — IgG4-TREAT and discloses speaker honoraria from Grifols, Teva, Biogen, Bayer, Roche, Novartis, Fresenius, travel funding from Merck, Grifols and Bayer and serving on advisory boards for Roche, Biogen and Alexion. K.H. was formerly employed by and holds stock or stock options in 23andMe, Inc. Murray.S. has in the past 3 years received consulting income from Acadia Pharmaceuticals, Aptinyx, atai Life Sciences, BigHealth, Biogen, Bionomics, BioXcel Therapeutics, Boehringer Ingelheim, Clexio, Delix

Therapeutics, Eisai, EmpowerPharm, Engrail Therapeutics, Janssen, Jazz Pharmaceuticals, NeuroTrauma Sciences, PureTech Health, Sage Therapeutics, Sumitomo Pharma, and Roche/Genentech. Dr. Stein has stock options in Oxeia Biopharmaceuticals and EpiVario. He has been paid for his editorial work on Depression and Anxiety (Editor-in-Chief), Biological Psychiatry (Deputy Editor), and UpToDate (Co-Editor-in-Chief for Psychiatry). He has also received research support from NIH, Department of Veterans Affairs, and the Department of Defense. He is on the scientific advisory board for the Brain and Behavior Research Foundation and the Anxiety and Depression Association of America. M.G.H. and J.J.G.M.V. are co-inventors on MuSK-related patents. LUMC, M.G.H., and J.J.G.M.V. receive royalties from these patents. LUMC receives royalties from a MuSK ELISA. M.G.H. is a consultant for argenx. Maike.S. has received speaker's honoraria and honoraria for attendance at advisory boards from argenx and Alexion. P.F. is employed by and holds stock or stock options in 23andMe, Inc. P.M. has been on the board of HealthNextGen. S.L. has received speaker's honoraria from Alexion, argenx, Hormosan and UCB and honoraria for attendance at advisory boards from Alexion, argenx, Biogen, HUMA, UCB and Roche. S.H. has received speaker's honoraria from Alexion, argenx, UCB and Roche and honoraria for attendance at advisory boards from Alexion, argenx and Roche. S.H. is a member of the medical advisory board of the German Myasthenia Society, DMG. M.R.T. reports trial support from argenx and Alexion, consultancies for argenx, HUMA and UCB Pharma and research funding from NMD Pharma, with all reimbursements received by Leiden University Medical Center. He is a member of the European Reference Network for Rare Neuromuscular Diseases (ERN EURO-NMD). The remaining authors declare no competing interests.

## Additional information

[1]Department of Psychiatry and Psychotherapy, Charité – Universitätsmedizin Berlin, corporate member of Freie Universität Berlin and Humboldt Universität zu Berlin, Berlin, Berlin, Germany. [2]Stanley Center for Psychiatric Research, Broad Institute of Harvard and MIT, Cambridge, Massachusetts, USA. [3]Department of Genetics, University of North Carolina at Chapel Hill, North Carolina, USA. [4]Department of Biological Sciences, Purdue University, West Lafayette, Indiana, USA. [5]Department of Psychiatry, Yale School of Medicine, West Haven, CT, USA. [6]Veterans Affairs Connecticut Healthcare Center, West Haven, CT, USA. [7]Vanderbilt Genetics Institute, Vanderbilt University Medical Center, Nashville, TN, USA. [8]Department of Biomedical Informatics, Vanderbilt University Medical Center, Nashville, TN, USA. [9]Department of Neurology with Experimental Neurology, Charité – Universitätsmedizin Berlin, corporate member of Freie Universität Berlin and Humboldt Universität zu Berlin, Berlin, Berlin, Germany. [10]Neuroscience Clinical Research Center, Charité – Universitätsmedizin Berlin, corporate member of Freie Universität Berlin and Humboldt Universität zu Berlin, Berlin, Berlin, Germany. [11]Department of Neurology, Beth Israel Deaconess Medical Center/Harvard Medical School, Boston, Massachusetts, USA. [12]Center for Stroke Research Berlin, Charité – Universitätsmedizin Berlin, corporate member of Freie Universität Berlin and Humboldt Universität zu Berlin, Berlin, Berlin, Germany. [13]Radcliffe Department of Medicine, University of Oxford, Oxford, United Kingdom. [14]Institute of Genomics, University of Tartu, Tartu, Estonia. [15]Department of Psychiatry, Icahn School of Medicine at Mount Sinai, New York, New York, USA. [16]Department of Genetics and Genomic Sciences, Icahn School of Medicine at Mount Sinai, New York, New York, USA. [17]Charles Bronfman Institute for Personalized Medicine, Icahn School of Medicine at Mount Sinai, New York, New York, USA. [18]Institute for Molecular Medicine FIMM, University of Helsinki, Helsinki, Finland. [19]deCODE Genetics/Amgen, Inc., Reykjavik, Iceland. [20]Department of Neurology, Kiel University, Kiel, Schleswig-Holstein, Germany. [21]Department of Psychiatry and School of Public Health, University of California San Diego, La Jolla, California, USA. [22]Veterans Affairs San Diego Healthcare System, Psychiatry Service, San Diego, California, USA. [23]23andMe, Inc., Sunnyvale, California, USA. [24]Leiden University Medical Center, Department of Neurology, Leiden, Zuid Holland, Netherlands. [25]Neuroimmunology, Kiel University, Institute of Clinical Chemistry, Kiel, Schleswig-Holstein, Germany. [26]Feinstein Institute for Medical Research, Northwell Health, Manhasset, New York, NY, USA. [27]Leiden University Medical Center, Department of Human Genetics, Leiden, Zuid Holland, Netherlands. [28]German Center for Mental Health (DZPG), partner site Berlin/Potsdam, Berlin, Germany. [29]These authors jointly supervised this work: Frauke Stascheit, Andreas Meisel, Stephan Ripke. ✉e-mail: sripke@broadinstitute.org

## Estonian Biobank Research Team

**Abdelrahman G. Elnahas[14], Kadri Reis[14] & Tõnu Esko[14]**

## 23andMe Research Team

**Pierre Fontanillas** ⓘ [23]

A full list of members and their affiliations appears in the Supplementary Information.

