## [Peer Review File · Nature Communications]

Genome-wide meta-analysis of myasthenia gravis uncovers new loci and provides insights into polygenic predictionREVIEWER COMMENTS

Reviewer #1 (Remarks to the Author):

Interesting paper. Well executed GWAS.

Quick comments:

- PRS: AUC is great to report, but how is class imbalance accounted for?
- Genetic correlations: there is some differential diagnosis in your cohorts, how does antibody versus EMR versus clinical diagnosis compare? Rare disease relatively so proxy cases might actually work pretty well.
- Summary stats: will they include 23andMe as well?

Nice job to the team.

Good luck,

Mike Nalls

Reviewer #2 (Remarks to the Author):

This is a very well executed GWAS that represents the largest genetic study of MG to date, tripling the sample size of prior studies. Especially nice that the authors aimed at incorporating all previously published data plus new data. The group has extensive expertise in conducting GWAS, and methods applied are state of the art (HLA and complement imputation, TWAS, LDSC, PRS). They also went at great length doing clinically meaningful subanalyses, like stratifying for antibody status and age at onset.

The results are displayed transparently, I like the Manhattans accompanying the tables with green arrows going up and red arrows going down, depending on the replication signal.

I do have a few (minor) comments:

1. Can the authors explain how to interpret the significance level of the complement analysis? In the HLA analysis they clearly show the different levels of significance, but with complement we only see ORs and 95% CI
2. I like the TWAS analysis but I wonder why most hits seem to cluster at chr 6? Could this be due to strange LD patterns at chr 6? And therefore: can we trust those chr 6 results in the TWAS? It is also surprising that so many different genes appear dysregulated by the GWAS hits on that chromosome. Also, wouldn't it be also interesting to do some pathway/GO analysis using the significant TWAS genes?
3. The authors used LDSC to study correlations with other traits, and those correlation make sense based on prior knowledge. I wonder if the authors also explored determining SNP based heritability using their data? The best performing PRS gives some indication, but is there a reason the authors do not report this?
4. The authors chose 12 traits to perform genetic correlation analysis. However, I would like to recommend to explore this a bit further. The autoimmune traits make sense of course. but I do not understand MS per se. I would also rather see traits like ALS/ Motor neuron disease, given the ATXN2 hit (although this might be a SNP and not a repeat variation), and since it is an important clinical mimic, and other traits that yield unexpected results (like epilepsy or Alzheimer's where immune processes also seem important). Not sure why the authors were so conservative in their choice of disorders, presumably to reduce bonferroni correction?
5. Did the authors consider exploring using the results to explore novel drugs for MG? For example using:
Sakaue, S. & Okada, Y. GREP: genome for REPositioning drugs. *Bioinformatics* 35, 3821-3823 (2019).
or:
Mirza, N. et al. Using common genetic variants to find drugs for common epilepsies. *Brain Commun.* 3, fcab287 (2021).

In summary: great work, maybe a few more (easy) add-on analyses will further enrich the paper.

Reviewer #3 (Remarks to the Author):

Summary:

Braun et al. conducted GWAS of Myasthenia gravis (MG), including 5,708 cases and 432,028 controls of European ancestry. They further conducted a replication study in 3,989 cases and

226,643 controls provided by 23andMe Inc. They identified 12 independent genome-wide significant index SNPs. They also conducted subgroup analyses in early-onset MG and late-onset MG. They further conducted HLA- and C4-focused analyses and some investigation in polygenic space (e.g. genetic correlation and PRS). Although I believe this would be a valuable resource for future studies, the investigation is very shallow and not complete. Please see my comments below.

Major comments:

1, What is the rationale behind including leave-one-out GWAS? The authors did not explain it, so it is not easy to follow the main text. This process can increase multiple testing burdens (12 cohorts, so 12 additional tests), but they did not change the genome-wide significance level. It seems not a reasonable strategy to me.

2, The authors should investigate more about the biological implications from 6 novel associations, which is one of the most important tasks in GWAS.

- Are they in LD with functional variants: missense variant, eQTL, sQTL, or variants in the regulatory region?

- Any pleiotropy with other autoimmune diseases?

3, HLA-oriented analysis is very intriguing. We usually do not observe opposite allelic effects between subgroups of the same autoimmune diseases. Do EOMG and LOMG have different autoantibody spectra? Any biological speculation on this point?

4, Did the authors care about the LD between C4 and HLA haplotypes? They need to show the results of conditional analyses to address this point. If the association is driven merely by LD, the results become meaningless.

5, In Figure 3, they tried to calculate the odds ratio compared with BS (BS is the reference in this analysis). Usually, researchers do multivariate regression analysis for this aim. A good example is Table 1 of Raychaudhuri et al. study (PMID: 22286218). They should try a more standard approach.

6, In TWAS analysis (Supplementary Table 10), the authors reported many genes in the MHC region. If the authors believe that the HLA alleles and C4 copy number variations causally contribute to MG development, they need to be cautious about reporting TWAS results in the HLA region. The very long-range LD structure in the HLA region produces many falsely positive TWAS results.

Minor comments:

1, Could the authors describe 12 cohorts in detail? Did the previous GWAS already analyze these cohorts? Or, any new cohorts or samples were added in this study?

2, How did the authors calculate the "effective N case", which appears multiple times on pages 7 and 8? There are several different ways to estimate these indices, so the authors need to clearly explain how they calculated it and provide a reference for it.

Response to referees

Dear Reviewers,

Thank you for the constructive feedback, which allowed us to address all comments and has significantly helped to improve our manuscript.

We used the following formatting hopefully enabling you find your own comments and our replies/changes immediately:

Black/Bold/Underscore: Your original comment/question/request

Blue: Our answer

Yellow highlight: Individual edits in response to your comment

Indented: Adapted or added paragraph from the former submission in response to your comment

Sincerely,

Alice Braun & Stephan Ripke

Reviewer #1 (Remarks to the Author):

- PRS: AUC is great to report, but how is class imbalance accounted for?

Thank you for this useful suggestion. Some of the datasets used for polygenic scoring are indeed very imbalanced with respect to the number of case and control subjects. We had reported R^2 on the liability scale in the supplementary tables. We now report them in the main text to show the impact of balancing case/control ratio. We revised the following paragraph from the main text, adapted parts are highlighted in yellow:

Results

Adapted paragraph:

“After excluding the test sample, our training GWAS comprised 5,318 cases and 431,304 controls. Conservatively adjusting for 50 tests resulted in a Bonferroni-corrected P -value of < 0.001 . In the test sample of 390 cases and 724 controls the polygenic risk score (PRS) for MG explained 4.21% of variation in disease status (R^2_{observed}) at a P -value thresholds (PT) of < 0.001 ($P = 5.12e^{-9}$, AUC = 0.607, $R^2_{\text{liability}} = 0.96\%$; Figure 3, Table S11). In a sample of 176 EOMG cases and 613 age-matched controls, the MG-PRS performed best at a PT < 0.1 . It explained 2.6% of the variation between early-onset cases and controls ($P = 2.58e^{-4}$, AUC = 0.579, $R^2_{\text{liability}} = 0.64\%$; Figure S15A, Table S12). In a sample of 213 LOMG cases and 62 matched controls, the MG-PRS explained 7.6% of the variation between cases and controls at PT < 0.001 ($P = 3.36e^{-4}$, AUC = 0.643, $R^2_{\text{liability}} = 1.75\%$; Figure S15B, Table S13).

We further assessed the performance of the MG-PRS when stratifying cases by antibody profile. We identified 73 cases that tested negative for AChR-Ab, including MuSK- and LRP4-Ab-positive patients. These cases were merged with 343 randomly selected controls. The MG-PRS explained 2.9% of the phenotypic variation at a PT of < 0.5 . However, the P -value remained above the Bonferroni-corrected threshold ($P = 7.4e^{-3}$, AUC = 0.556, $R^2_{\text{liability}} = 0.71\%$; Figure S16A, Table S14). Finally, we calculated the MG-PRS in 371 AChR-Ab-positive cases and 333 independent controls which explained 4.8% of phenotypic variation **at a PT < 0.0001** ($P = 1.18e^{-6}$, AUC = 0.620, $R^2_{\text{liability}} = 1.01\%$; Figure S16B, Table S15).”

- Genetic correlations: there is some differential diagnosis in your cohorts. how does antibody versus EMR versus clinical diagnosis compare? Rare disease relatively so proxy cases might actually work pretty well.

We have now explored the genetic correlation between these three data sources. As expected, the genetic correlation between AChR antibody positive cases and clinical cases is the highest due to more strict inclusion criteria. All Correlations between the data sources are nominally significant.

p1	p2	rg	se	p
EHR	Clinic mixed Ab	0.4774	0.2430	0.0494
EHR	Clinic AChR Ab	0.4021	0.1323	0.0024
Clinic mixed Ab	Clinic AChR Ab	0.6831	0.2137	0.0014

We have added the following paragraphs to the main text:

Results

Newly added paragraph:

“We have further assessed the genetic correlation between datasets ascertained through clinical studies without antibody-based inclusion criteria (effective $n_{\text{half}} = 507$), clinical studies limited to AChR-ab cases (effective $n_{\text{half}} = 5,067$) and electronic healthcare records (EHR) (effective $n_{\text{half}} = 4,293$). We observe the highest correlation between AChR-ab filtered and clinical datasets ($r_g = 0.6831$, $SE = 0.2137$, $P = 0.0014$). However, correlations between EHR based and clinical samples ($r_g = 0.4774$, $SE = 0.2430$, $P = 0.0494$) and between EHR based and AChR-ab filtered ($r_g = 0.4021$, $SE = 0.1323$, $P = 0.0024$) were nominally significant.”

Discussion

Newly added paragraph:

“We detected significant genetic correlation among all data sources, EHR, clinical samples, and clinical samples limited to AChR-ab positive cases. As would be expected, the strongest correlation was found between the more rigorously collected clinical samples and AChR-ab positive cases, underscoring some variability introduced by the EHR collections.”

- Summary stats: will they include 23andMe as well?

For the current publication we have only requested the replication of a limited number of SNPs through 23andMe.

While all reported genome-wide significant hits are included in the results table 1, we never acquired access to whole genome 23andme results and these are therefore not included in whole genome summary statistics that we will share via GWAS Catalog.

Reviewer #2 (Remarks to the Author):

1. Can the authors explain how to interpret the significance level of the complement analysis? In the HLA analysis they clearly show the different levels of significance, but with complement we only see ORs and 95% CI

In response to another reviewer's feedback, we have thoroughly revised this sub-analysis and identified several issues with the original approach. Consequently, the updated results diverge from those initially presented. Additionally, we have transitioned from using a graph to presenting the data in a simplified table format, which we believe will be adequately informative.

These are the revised paragraphs and results:

Results

Adapted paragraph:

"We performed association tests and meta-analyses of imputed complement component 4 (C4) haplotypes in 1,927 MG cases and 5,549 controls, 1,080 EOMG cases and 3,321 controls, and 846 LOMG cases and 2,179 controls. Two isoforms of C4 are encoded by the genes C4A and C4B located at the MHC class III region. These vary in size and copy number combinations, resulting in long forms (L; ~21 kilobases) and short forms (S; ~14 kilobases). Specifically, we examined four common structural haplogroups of C4A (A) and C4B (B) (BS, AL-BS, AL-BL, and AL-AL). We calculated multiple logistic regression models with BS as the reference haplogroup, which has the fewest gene copy numbers. Resulting odds ratios ranged from 0.46-0.54 in MG, 0.21-0.27 in EOMG, and 1.83-2.27 in LOMG (Table S7)."

Methods

Adapted paragraph:

"We used a reference panel based on 1,265 sequenced individuals from the Genomic Psychiatry Cohort to infer C4 haplotypes (Kamitaki et al., 2020), which were imputed using Beagle 5.457 (Browning et al., 2018). We initially filtered the imputed C4 variants based on the gene composition levels (e.g., BS, AL-BS, AL-BL, AL-AL) by allele frequency (> 0.01) in controls and imputation quality (INFO > 0.80). Subsequently, we removed individuals with dosage sums ≤ 1.9 to avoid these being attributed to the reference group in the logistic regression model. Dosages for all remaining C4 variants were then modeled together in a logistic regression for each outcome, incorporating the first six principal components and excluding the BS reference group, denoting the shortest C4 variant. Fixed-effects inverse-variance weighted meta-analyses for MG, EOMG, and LOMG were conducted in R version 4.3.2 (R Core Team, 2023) using meta version 7.0-0 (Balduzzi et al., 2019)."

Phenotype	C4 gene composition	OR	95% CI_Lower	95% CI_Upper	P-value
MG	AL_BL	0.5175687	0.4478411	0.5981526	4.63E-19
MG	AL_BS	0.5432198	0.4638815	0.6361275	3.58E-14
MG	AL_AL	0.4560478	0.3725808	0.5582134	2.68E-14
EOMG	AL_BL	0.273188	0.2257446	0.3306021	1.50E-40
EOMG	AL_BS	0.2389753	0.192345	0.2969103	3.28E-38
EOMG	AL_AL	0.2110694	0.1591927	0.2798513	3.13E-27
LOMG	AL_BL	1.952876	1.442282	2.644229	1.50E-05
LOMG	AL_BS	2.26632	1.659156	3.095673	2.72E-07
LOMG	AL_AL	1.827736	1.269427	2.631595	1.18E-03

2. I like the TWAS analysis but I wonder why most hits seem to cluster at chr 6? Could this be due to strange LD patterns at chr 6? And therefore: can we trust those chr 6 results in the TWAS? It is also surprising that so many different genes appear dysregulated by the GWAS hits on that chromosome. Also, wouldn't it be also interesting to do some pathway/GO analysis using the significant TWAS genes?

Based on your feedback and the comment of another reviewer we have now excluded the extended MHC region (25–35Mb) from the TWAS altogether. Initial analyses indeed highlighted MHC genes potentially in strong LD, and we have conducted MHC-specific analyses elsewhere.

Results

Adapted paragraph:

“TWAS using the individual tissue-based prediction matrix of gene expression identified 138 significant genes whose transcript expression was significantly associated with MG. Permutation and co-localization tests identified 56 and 45 gene hits below the significance threshold. Combining TWAS, permutation, and co-localization tests, we identified 24 significant unique genes with high confidence. The TWAS highlighted individual genes in the highly complex loci on chromosome 12 and 17, as well as genes in additional loci that did not reach genome-wide significance in the GWAS, including *PAPPA* and *EPS15L1* (Table S10, Figure S12-14).”

Methods

Adapted paragraph:

“Transcriptome-wide association study (TWAS) was performed using FUSION (Gusev et al., 2016) to predict tissue-specific gene expression based on the MG discovery GWAS summary statistics excluding the extended MHC region (Chromosome 6: 25-35Mb). Gene expression weights from three Genotype-Tissue Expression (GTEx) version 8 (The GTEx Consortium, 2020) tissues were used for transcriptomic imputation and association testing: Muscle Skeletal ($n = 8,602$), Nerve Tibial ($n = 11,360$), and Whole Blood ($n = 8,059$). Gene associations were considered significant at a P -value $< 0.05/N$ of genes per tissue. A permutation test, which shuffles the quantitative trait loci (QTL) weights ($N_{\max} = 1000$) was performed to correct inflated associations from by-chance QTL co-localization. Co-localization was computed for genes with a P -value $< 1e^{-5}$. A posterior probability value ≥ 0.75 was considered as evidence for the expression-QTL-GWAS pair influencing both the expression and the GWAS trait in a particular region.”

3. The authors used LDSC to study correlations with other traits, and those correlation make sense based on prior knowledge. I wonder if the authors also explored determining SNP based heritability using their data? The best performing PRS gives some indication, but is there a reason the authors do not report this?

Thank you for pointing out the lack of these metrics in our analyses. We have now added them to the main text:

Results

Newly added paragraph:

“We have estimated the SNP-based heritability (h^2_{SNP}) of MG using LDSC (Finucane et al., 2015) and the genome-wide complex trait analysis (GCTA) software package (Yang et al., 2011). The h^2_{SNP} on the liability scale based on the 7,476 clinically ascertained individuals with available genotypes measured through GCTA was 0.1818 (SE = 0.0149). The h^2_{SNP} for the same 7,476 individuals was 0.1184 (SE = 0.0272) using LDSC. The h^2_{SNP} on the liability scale as measured by LDSC based on the summary statistics of the whole sample of 437,736 individuals including biobank-based summary statistics was 0.0552 (SE = 0.0091).”

Discussion

Newly added paragraph:

“We present h^2_{SNP} estimates ranging from 0.056 to 0.18 on the liability scale, obtained using LDSC and GCTA GREML. Our GCTA estimate in clinically ascertained samples ($n = 7,476$) was similar to the value originally reported by Renton and colleagues (2015) ($h^2 = 0.26$). Our LDSC estimate in the same sample was lower, which was expected given that LDSC h^2 estimates tend to be smaller than those from GCTA and capture the lower bound of h^2 (Evans et al., 2018). Our LDSC h^2 estimate (0.0552) was lower still when including samples from large biobanks. This may be due to a higher rate of misdiagnosis in biobank cases, and/or due to clinically ascertained cases and controls representing a more severe phenotype (Schork et al., 2019).”

Methods

Newly added paragraph:

"We used LDSC (Finucane et al., 2015) and the genome-based restricted maximum likelihood (single-component GREML) module implemented in GCTA (Yang et al., 2011) to assess h^2_{SNP} of MG in our sample. We used summary statistics including all samples in the discovery GWAS to estimate h^2_{SNP} via LDSC and a merged the PLINK files of all available genotypes as input for GCTA along with the first six PCs. We assumed a population prevalence of 0.0002 to transform the heritability estimates to the liability scale."

4. The authors chose 12 traits to perform genetic correlation analysis. However, I would like to recommend to explore this a bit further. The autoimmune traits make sense ofcourse, but I do not understand MS per se. I would also rather see traits like ALS/ Motor neuron disease, given the ATXN2 hit (although this might be a SNP and not a repeat variation), and since it is an important clinical mimic, and other traits that yield unexpected results (like epilepsy or Alzheimer's where immune processes also seem important). Not sure why the authors were so conservative in their choice of disorders, presumable to reduce bonferroni correction?

Thank you for this suggestion. We have indeed relied only on pre-selected traits in order to reduce the multiple testing burden. We have adapted our pre-selected analysis to include epilepsy, Alzheimer's disease, and ALS. Furthermore, we have run genetic correlation analyses with 1,445 medical phecodes in the UK Biobank.

We have added/ adapted the following paragraphs to the main text:

Results

Newly added paragraph:

"Genetic correlation analyses performed with 1,445 medical phecodes in the UK biobank and MG highlighted eight traits that passed the Bonferroni adjusted P -value threshold of $3.42e^{-5}$ (correcting for overall 1,460 tests). We found the strongest correlations between MG and pernicious anemia ($r_g = 0.8216$, $P = 2.95e^{-8}$, $SE = 0.1482$), rheumatoid arthritis and other inflammatory polyarthropathies ($r_g = 0.522$, $P = 1.36e^{-5}$, $SE = 0.12$) and thyrotoxicosis with or without goiter ($r_g = 0.4925$, $P = 9.73e^{-8}$, $SE = 0.0924$). Results for all traits below nominal significance ($P < 0.05$) are presented in Table S19."

Discussion

Adapted paragraph:

"In an analysis of pre-selected traits we replicate the genetic correlation of MG with type 1 diabetes, rheumatoid arthritis, and vitiligo and further found positive correlations of MG with systemic lupus erythematosus and multiple sclerosis. LOMG was most strongly associated with vitiligo and EOMG showed a strong association with systemic sclerosis. In a less restrictive analysis of the UK Biobank dataset, pernicious anemia and thyrotoxicosis emerged as the strongest correlations, and further confirmed the correlation with rheumatoid arthritis. However, these correlations could be driven by the presence of autoimmune comorbidity. Future studies could further investigate the results, e.g. by excluding co-morbid individuals."

Methods

Adapted paragraph:

"The bivariate genetic correlations of MG, EOMG, and LOMG with 15 pre-selected neurological and autoimmune traits were computed via LDSC (Bulik-Sullivan et al., 2015). We downloaded publicly available summary statistics from the GWAS catalog website [<https://www.ebi.ac.uk/gwas/>] and formatted summary statistics from Dr. Alkes Price's research group's repository [https://alkesgroup.broadinstitute.org/sumstats_formatted]. We have additionally performed genetic correlation analyses between MG and 1,445 medical phecodes, utilizing data extracted from primary care records, hospitalizations, cancer registers, and death records of 500,000 participants from the UK Biobank described in detail by Pietzner and colleagues (2024). We have excluded the UK Biobank sample from our summary statistics for this purpose. "

We have adapted Tables S16-18, Figure S17 and added Table S19 accordingly.

5. Did the authors consider exploring using the results to explore novel drugs for MG? For example using: Sakaue, S. & Okada, Y. GREP: genome for REPositioning drugs. Bioinformatics 35, 3821-3823 (2019). or: Mirza, N. et al. Using common genetic variants to find drugs for common epilepsies. Brain Commun. 3, fcab287 (2021).

Thank you again for this helpful suggestion, which we have not explored in the initial manuscript. We have run the GREP pipeline for ATC and ICD disease categories on all genes in the proximity of the GWAS index SNP prioritized via the FUMA SNP2GENE pipeline.

Results

Newly added paragraph:

“We performed drug target enrichment analysis on the set of 52 prioritized genes as input to the Genome for REPositioning drugs (GREP) pipeline (Sakaue & Okada, 2019). Significant enrichment was observed only for the ATC category “ectoparasiticides, incl. scabicides, insecticides and repellents” (OR = 28.73 and Fisher’s exact $P = 4.3e^{-2}$). However, this result does not withstand multiple testing correction, and the target gene *ALDH2* is situated within the complex chromosome 12 locus.”

Methods

Newly added paragraph:

“To evaluate the enrichment of clinical indication categories (ICD-10 and ATC) of druggable target genes in our GWAS, we utilized the set of genes prioritized by FUMA as input to the GREP pipeline (Sakaue & Okada, 2019). The pipeline employs Fisher’s exact tests to determine whether the gene set demonstrates enrichment in genes targeted by medications.”

Reviewer #3 (Remarks to the Author):

Major comments:

1. What is the rationale behind including leave-one-out GWAS? The authors did not explain it, so it is not easy to follow the main text. This process can increase multiple testing burdens (12 cohorts, so 12 additional tests), but they did not change the genome-wide significance level. It seems not a reasonable strategy to me.

Thank you for highlighting this critical point, which we did not express clearly. We conducted a genome-wide leave-one-out analysis because we noticed early in the process that combining samples from various data sources introduced heterogeneity, leading to inconsistent associations across clinical, electronic healthcare, and autoantibody-based datasets.

We observed that some associations were lost, just barely failing genome-wide significance, with the addition of new cohorts. Consequently, we decided not to exclude these associations from our planned replication effort with 23andMe. We also noted that if we applied a Bonferroni correction including 12 additional tests ($P < 3.85e^{-9}$), three index SNPs would no longer meet the significance threshold, and we now indicated this in table 1.

Table 1A. Discovery GWAS, replication and combined meta-analysis results for MG

Nearest gene	Index SNP	CHR	BP (GRCh37)	A1/A2	FRQ case	FRQ control	Proxy	Discovery			Replication			Discovery and replication		
								P	OR	SE	P-rep	OR-rep	SE-rep	P-comb	OR-comb	SE-comb
PTPN22	rs2476601 ^b	1	114377568	A/G	0.121	0.13	same	2.77e⁻³¹	1.45965	0.0325	3.92e ⁻¹⁴	1.311637	0.035159	3.25e⁻⁴³	1.38944	0.0239
MAGI3	rs7522138 ^d	1	114217705	A/G	0.441	0.47	same	6.77e⁻⁴⁹	0.88462	0.0212	2.91e ⁻⁰²	0.957836	0.022749	3.36e⁻⁴⁸	0.91796	0.0155
CHRNA1	rs6433501 ^{a,b}	2	175616667	G/A	0.151	0.112	same	3.54e ⁻⁰⁷	1.17081	0.031	1.01e ⁻⁰⁵	1.14714	0.03172	2.69e⁻¹¹	1.15917	0.0222
CTLA4	rs231779 ^b	2	204734487	T/C	0.386	0.451	rs231770	1.24e ⁻⁰⁶	1.11149	0.0218	2.37e ⁻⁰⁵	1.09845	0.024511	6.87e⁻¹⁰	1.10572	0.0163
TNIP1	rs6861227 ^{a,c,d}	5	150447128	G/T	0.165	0.128	same	7.44e ⁻⁰⁸	1.16754	0.0288	2.07e ⁻⁰²	1.075955	0.031392	3.03e⁻⁴⁸	1.12479	0.0212
MHC	rs1264706 ^b	6	30063652	C/G	0.129	0.072	same	1.26e⁻³⁴	1.61996	0.0393	1.92e ⁻⁰²	1.091196	0.041711	3.44e⁻²⁵	1.34528	0.0286
TBX18	rs215918	6	85513783	A/G	0.405	0.374	same	3.42e⁻⁴⁸	0.88754	0.0216	3.30e ⁻⁰³	0.938976	0.02322	3.84e⁻⁴⁹	0.91101	0.0158
RNASET2	rs2301436 ^d	6	167437988	T/C	0.493	0.477	same	1.09e ⁻⁰⁷	1.11818	0.021	6.69e ⁻⁰³	1.05781	0.022716	2.33e⁻⁴⁸	1.08992	0.0154
FAM76B	rs4409785 ^b	11	95311422	C/T	0.192	0.169	same	2.73e⁻¹¹	1.19494	0.0267	3.44e ⁻⁰³	1.08206	0.028953	1.48e⁻¹¹	1.14168	0.0196
ATXN2	rs4766578 ^a	12	111904371	T/A	0.516	0.406	same	6.72e ⁻⁰⁸	1.12345	0.0216	2.09e ⁻⁰⁴	1.08362	0.022773	2.35e⁻¹⁰	1.1044	0.0157
IKZF3	rs12946510	17	37912377	T/C	0.476	0.493	same	7.61e⁻¹¹	1.1474	0.0211	2.38e ⁻⁰⁴	1.08288	0.022772	8.28e⁻¹³	1.11717	0.0155
TNFRSF11A	rs7239261 ^a	18	60005046	A/C	0.457	0.436	same	3.98e⁻²⁰	1.21458	0.0212	4.43e ⁻⁰⁴	1.079214	0.022905	2.46e⁻¹⁹	1.15016	0.0156

Table 1B. Discovery GWAS, replication and combined meta-analysis results for EOMG

Nearest gene	Index SNP	CHR	BP (GRCh37)	A1/A2	FRQ case	FRQ control	Proxy	P	OR	SE	P-rep	OR-rep	SE-rep	P-comb	OR-comb	SE-comb
PTPN22	rs2476601 ^b	1	114377568	A/G	0.156	0.136	same	3.61e⁻⁴⁶	1.46976	0.0653	2.77e ⁻⁰⁴	1.30479	0.074616	1.15e⁻¹¹	1.39585	0.0491
MHC	rs2853986 ^b	6	31338844	T/C	0.682	0.907	same	4.06e⁻¹⁰⁵	0.2395	0.0656	1.28e ⁻¹¹	0.636760	0.064406	2.67e⁻⁸¹	0.39404	0.046

Table 1C. Discovery GWAS, replication and combined meta-analysis results for LOMG

Nearest gene	Index SNP	CHR	BP (GRCh37)	A1/A2	FRQ case	FRQ control	Proxy	P	OR	SE	P-rep	OR-rep	SE-rep	P-comb	OR-comb	SE-comb
PTPN22	rs2476601 ^b	1	114377568	A/G	0.141	0.132	same	9.26e⁻¹⁵	1.47669	0.0503	3.92e ⁻¹⁴	1.311637	0.035159	5.11e⁻²⁷	1.3637	0.0288
CTLA4	rs231779 ^{b,d}	2	204734487	C/T	0.572	0.515	rs231770	2.66e ⁻⁰⁶	0.85172	0.0342	2.37e ⁻⁰⁵	0.910374	0.024511	4.99e⁻⁴⁹	0.89003	0.0199
MHC	rs72848204 ^b	6	32594073	G/T	0.749	0.697	same	1.20e⁻²³	1.50953	0.0411	2.37e ⁻⁰²	1.049218	0.024296	1.04e⁻¹¹	1.15281	0.0209
TNFRSF11A	rs7239261 ^a	18	60005046	A/C	0.492	0.368	same	1.56e⁻¹⁴	1.29576	0.0337	4.43e ⁻⁰⁴	1.079214	0.022905	1.50e⁻¹²	1.14339	0.0189

Abbreviations: BP (GRCh37), Base Pair position on Genome Reference Consortium Human Build 37; CHR, chromosome; comb, combined meta-analysis of discovery and control GWAS results; FRQ, frequency.

Footnotes: ^aReached genome-wide significance in leave-one-out discovery GWAS; ^bPreviously published as genome-wide significant locus; ^cOnly previously reported in EOMG.

^dWould not withstand a multiple testing correction for 13 million tests ($P < 3.85e^{-9}$), accounting for the leave-one-out GWAS.

Methods

Adapted paragraph:

We conducted leave-one-out GWAS to identify additional significant candidate loci for replication that were potentially lost due to heterogeneity in our overall sample. All newly identified or previously reported GWAS loci were subsequently confirmed or rejected via replication analyses, including meta-analysis and binomial sign-tests.

2. The authors should investigate more about the biological implications from 6 novel associations, which is one of the most important tasks in GWAS.

- Are they in LD with functional variants: missense variant, eQTL, sQTL, or variants in the regulatory region?

Thank you for bringing the absence of functional characterization of the identified loci to our attention. In response to your comment, we have conducted a gene prioritization analysis and functional annotation using the FUMA SNP2gene pipeline and included a discussion on genes and nonsynonymous variants highlighted by this analysis. For completeness, we considered all loci in our study, including those previously reported.

additions to the main manuscript:

Results

Newly added paragraph:

“To link loci implicated by the GWAS to protein coding genes, we applied positional, expression quantitative trait loci (eQTL), and chromatin interaction gene mapping. This resulted in 52 mapped genes across 11 loci, excluding the extended MHC region. Four of these loci (rs6433501, rs231779, rs6861227, rs215918) only contained a single protein-coding gene (*CHRNA1*, *CTLA4*, *TNIP1*, *TBX18*). Furthermore, our analyses highlighted the index SNP rs2476601 in *PTPN22* on chromosome 1 (PIP = 66%) and rs231775 in *CTLA4* on chromosome 2 (PIP = 14%; max. PIP = 19%; LD- r^2 = 0.97 with lead variant rs231779) as nonsynonymous variants. On chromosome 11 there were no protein-coding genes within the defined locus boundaries. For the remaining loci, positional, eQTL, and chromatin interaction mapping nominated multiple genes in the loci, none of which can be prioritized with high confidence. Gene-level results are shown in Table S8, annotated results for SNPs in the credible set with PIP > 0.01 are included in Table S9.”

Discussion

Newly added paragraph:

“We employed multiple mapping techniques to associate GWAS-identified loci with protein-coding genes, identifying 52 mapped genes across 11 loci, excluding the MHC region, which we analyzed separately. Notably, four loci contained only a single gene. First, *CHRNA1*, a nicotinic acetylcholine receptor subunit and a target of neuromuscular blocking agents (Ryaboshapkina & Hammar, 2019). Second, *CTLA4*, a member of the CD28 immunoglobulin superfamily regulating T-cell responses and an immunotherapy target (Van Coillie et al., 2020). Third, *TNIP1*, which plays a key role in inflammatory processes and showed a high loss-of-function intolerant probability (pLI) of 0.93. Fourth, *TBX18*, involved in embryonic development (Sheeba et al., 2017), which had a high pLI score of > 0.99 and mapped SNPs reached a maximum CADD score of 18.77, indicating deleteriousness (Kircher et al., 2014; Amendola et al., 2015). Furthermore, our analyses highlighted rs2476601 in *PTPN22* on chromosome 1 and rs231775 in *CTLA4* on chromosome 2 as nonsynonymous variants.”

Methods

Newly added paragraph:

“In order to map SNPs within the identified loci to protein coding genes we applied the SNP2GENE module implemented in FUMA version 1.5.2 (Watanabe et al., 2017) via positional (10 kilobase window), eQTL (GTEx version 8 tissues, database of immune cell expression eQTLs), and chromatin interaction mapping (HiC of adult and fetal cortex, and GSE87112 tissues). We used the discovery GWAS summary statistics along with the set of 11 pre-defined index SNPs as input files for FUMA. Due to its complex LD structure we excluded the extended MHC region (Chromosome: 6 25–35Mb) from all analyses. We additionally filtered the mapped genes with the LD-based locus boundaries defined by RICOPIILI (LD- r^2 > 0.1). Furthermore, we intersected FUMA functional annotations with SNPs in the 95% credible set with a PIP > 0.01 (calculated via the R package coloc 5.2.3 finemap.abf module (Wallace, 2021)).“

- Any pleiotropy with other autoimmune diseases?

We had included a GWAS Catalog lookup in the original manuscript. We have, however, not referred to it in the main text. We have adapted that now and added an additional lookup for each of the index variants across several repositories. We have added the following paragraphs to the manuscript:

Results

Newly added paragraph:

“We conducted a lookup of index SNPs included in the discovery GWAS and those in LD ($LD-r^2 > 0.1$) in GWAS Catalog to assess pleiotropy. Furthermore, GWAS Catalog, UK Biobank, and FinnGen R6 datasets were searched for associations with individual index SNPs. The lookups revealed large pleiotropy with other autoimmune diseases and immune-related molecular phenotypes, including white blood cell count. GWAS Catalog results for all SNPs with $P < 1e^{-4}$ are included in Table S1. Genome-wide significant associations with other traits for each index SNP are shown in Figure S18.”

Methods

Newly added paragraph:

“In order to assess pleiotropy, we conducted a lookup of all SNPs in the discovery GWAS and those in LD ($LD-r^2 > 0.1$) in GWAS Catalog (version from September 2018) (Sollis et al., 2023) for associations with other traits. Additionally, we performed a lookup for the 11 individual index SNPs highlighted by the combined meta-analysis, excluding the MHC in data sources aggregated by Open Targets Genetics. The data sources include associations identified by the SAIGE study and the Neale lab conducted in the UK Biobank, summary statistics from GWAS Catalog, and FinnGen Release 6 (Ghoussaini et al., 2021).”

3. HLA-oriented analysis is very intriguing. We usually do not observe opposite allelic effects between subgroups of the same autoimmune diseases. Do EOMG and LOMG have different autoantibody spectra? Any biological speculation on this point?

The vast majority of cases in this analysis tested positive for AChR antibodies as most clinical samples used in this included were restricted to this antibody profile. Of our EOMG cases, ~97% were AChR antibody-positive. Of our LOMG cases, ~96% were AChR antibody-positive. It is thus unlikely to significantly impact the age-of-onset specific analyses. Based on our results, HLA alleles may indeed play a role in modulating the age of onset variation in MG. However, it's crucial to emphasize the necessity of replicating these findings in a large, independent sample to ensure robustness and generalizability. We have now phrased the discussion more carefully.

Discussion

Adapted paragraph:

“The MHC represents one of the strongest signals in MG across subtypes. Our analyses identified *HLA-B*08:01* as the top associated allele, replicating the results of smaller-scale studies. While we confirm a protective effect of *HLA-DQB1*03* and a risk-conferring effect of *HLA-DRB1*07* on LOMG we found opposite direction effects in EOMG. Inverse odds ratios based on the onset subtype were additionally observed for *HLA-B*08:01* and *DRB1*03:01*. Given that the vast majority of cases in this analysis tested positive for AChR-Ab (~97% of EOMG and ~96% of LOMG cases), it is unlikely that antibodies drive these effects. These findings could imply a modulating effect of HLA alleles on the time of disease onset. However, due to the complex LD structure of the MHC, independent large-scale sequencing studies are needed to dissect the functionality of individual HLA alleles in the etiology of MG.”

5. In Figure 3, they tried to calculate the odds ratio compared with BS (BS is the reference in this analysis). Usually, researchers do multivariate regression analysis for this aim. A good example is Table 1 of Raychaudhuri et al. study (PMID: 22286218). They should try a more standard approach.

Thank you for this suggestion, which additionally made us aware of flaws with the initially presented sub-analysis, including a sign error. We have now completely redone this analysis using the dosages for all C4 haplotypes in a single regression model excluding the reference as proposed in your comment. We reached out to the authors of the publications you referenced to verify our statistical model.

Results

Adapted paragraph:

“We performed association tests and meta-analyses of imputed complement component 4 (C4) haplotypes in 1,927 MG cases and 5,549 controls, 1,080 EOMG cases and 3,321 controls, and 846 LOMG cases and 2,179 controls. Two isoforms of C4 are encoded by the genes C4A and C4B located at the MHC class III region. These vary in size and copy number combinations, resulting in long forms (L; ~21 kilobases) and short forms (S; ~14 kilobases). Specifically, we examined four common structural haplogroups of C4A (A) and C4B (B) (BS, AL-BS, AL-BL, and AL-AL). We calculated multiple logistic regression models with BS as

the reference haplogroup, which has the fewest gene copy numbers. Resulting odds ratios ranged from 0.46-0.54 in MG, 0.21-0.27 in EOMG, and 1.83-2.27 in LOMG (Table S7).”

Methods

Adapted paragraph:

“We used a reference panel based on 1,265 sequenced individuals from the Genomic Psychiatry Cohort to infer C4 haplotypes (Kamitaki et al., 2020), which were imputed using Beagle 5.457 (Browning et al., 2018). We initially filtered the imputed C4 variants based on the gene composition levels (e.g., BS, AL-BS, AL-BL, AL-AL) by allele frequency (> 0.01) in controls and imputation quality (INFO > 0.80). Subsequently, we removed individuals with dosage sums ≤ 1.9 to avoid these being attributed to the reference group in the logistic regression model. Dosages for all remaining C4 variants were then modeled together in a logistic regression for each outcome, incorporating the first six principal components and excluding the BS reference group, denoting the shortest C4 variant. Fixed-effects inverse-variance weighted meta-analyses for MG, EOMG, and LOMG were conducted in R version 4.3.2 (R Core Team, 2023) using meta version 7.0-0 (Balduzzi et al., 2019).”

Phenotype	C4 gene composition	OR	95% CI_Lower	95% CI_Upper	P-value
MG	AL_BL	0.5175687	0.4478411	0.5981526	4.63E-19
MG	AL_BS	0.5432198	0.4638815	0.6361275	3.58E-14
MG	AL_AL	0.4560478	0.3725808	0.5582134	2.68E-14
EOMG	AL_BL	0.273188	0.2257446	0.3306021	1.50E-40
EOMG	AL_BS	0.2389753	0.192345	0.2969103	3.28E-38
EOMG	AL_AL	0.2110694	0.1591927	0.2798513	3.13E-27
LOMG	AL_BL	1.952876	1.442282	2.644229	1.50E-05
LOMG	AL_BS	2.26632	1.659156	3.095673	2.72E-07
LOMG	AL_AL	1.827736	1.269427	2.631595	1.18E-03

4, Did the authors care about the LD between C4 and HLA haplotypes? They need to show the results of conditional analyses to address this point. If the association is driven merely by LD, the results become meaningless.

Thank you for highlighting this issue. We used two different imputation reference panels for the HLA and C4 analyses which made it challenging to directly conduct conditional analyses. We have now addressed this issue by calculating the conditioning on C4 expression in the HLA association analyses. The C4 expression was calculated based on the imputed C4 haplotypes based on the following formula published by Sekat et al. (PMID: 26814963):

$C4A$ expression = (0.47 * AL) + (0.47 * AS) + (0.20 * BL); $C4B$ expression = (1.03 * BL) + (0.88 * BS).

We have adapted the following paragraphs to the main manuscript:

Results

Adapted paragraph:

“We performed association analyses of 135 imputed HLA class I and II alleles in a subsample of 1,927 cases and 5,549 controls with available genotypes. Subsequently, we conducted separate analyses for EOMG (1,080 cases and 3,321 controls) and LOMG (846 cases and 2,179 controls). Effects sizes for the strongest associations across the three subgroups are visualized in Figure 2.

Our analysis revealed *HLA-B*08:01* as the top risk-conferring HLA allele for MG (OR = 2.349, $P = 1.15e^{-52}$, SE = 0.056; Table S4, Figure S9). The top protective allele was *HLA-C*05:01*, (OR = 0.599, $P = 3.21e^{-10}$, SE = 0.082). After conditioning on the top variant rs4143332 (LD- r^2 of 0.988 to *HLA-B*08:01*), or *C4* expression levels *HLA-C*05:01* remained nominally significant (rs4143332: OR = 0.6869, $P = 5.662e^{-6}$, SE = 0.0828; *C4* expression: OR = 0.68991 SE = 0.0865, $P = 1.774e^{-5}$), indicating partial independence from *HLA-B*08:01* and *C4*. After conditioning on *C4A* and *C6B* expression levels *HLA-B*08:01* remained genome-wide significant (OR = 2.34527, SE = 0.0777, $P = 5.69e^{-28}$).

*HLA-B*08:01* was also the top risk-conferring allele associated with EOMG (OR = 4.677, $P = 2.18e^{-94}$, SE = 0.075; Table S5, Figure S10) while *DRB1*07:01* was the top protective (OR = 0.3996, $P = 1.11e^{-16}$, SE = 0.111). After conditioning on the top variant rs2596565 (LD- r^2 of 1 to *HLA-B*08:01*), the protective effect of *HLA-DRB1*07:01* remained genome-wide significant (OR = 0.529, $P = 2.63e^{-8}$, SE = 0.115), indicating *HLA-DRB1* as an independent HLA-haplotype. When additionally conditioning on *C4* expression both effects remained genome-wide significant (*HLA-B*08:01*: OR = 5.00531 SE = 0.0.1082 $P = 4.453e^{-50}$; *DRB1*07:01*: OR = 0.34311, SE = 0.1334, $P = 1.048e^{-15}$).

The top HLA allele associated with LOMG was *HLA-DRB1*03:01* which implied a protective effect (OR = 0.479, $P = 2.37e^{-9}$, SE = 0.123; Table S6, Figure S11). After conditioning on *C4* expression the effect remained genome-wide significant (OR = 0.36400, SE = 0.1583 $P = 1.724e^{-10}$). We observed a significant opposite direction effect for *HLA-DRB1*03:01* in MG (OR = 1.916, $P = 2.72e^{-30}$, SE = 0.057) and much more pronounced in EOMG (OR = 3.689, $P = 6.12e^{-71}$, SE = 0.073). The top risk-conferring allele *DRB1*07:01* in LOMG did not reach genome-wide significance (OR = 1.414, $P = 8.86e^{-5}$, SE = 0.088).“

Discussion

Adapted paragraph

“Our analyses have multiple limitations to be considered. Firstly, antibody phenotypes were not available for all cohorts. A GWAS stratified by antibody profiles could reveal distinct etiological differences, considering the significant impact of antibody subtypes on therapeutic stratification, future studies should concentrate on the currently low number of genotyped AChR-Ab negative MG patients. The diagnoses in the replication sample were based on self-reporting, which could potentially render them less reliable compared to diagnoses obtained through clinical ascertainment. Furthermore, conditional analyses conducted by employing *C4* gene expression may not fully account for the LD with HLA alleles, which were imputed using a different reference panel. Sequencing studies are needed to confirm the observed effects and establish their independence. Finally, our analyses are based on European ancestry samples, limiting the discovery rate and the predictive capability of PRS across ancestries. In the next step, we aim to expand our analyses to include samples from non-European ancestries.“

Methods

Adapted paragraph:

“For all available genotypes, we imputed HLA alleles via a European 1000 Genomes phase 3 reference panel of 503 individuals with HLA types inferred from sequencing data. The reference was downloaded from the CookHLA GitHub repository and includes 151 HLA alleles with a frequency > 0.01. Pre-phasing and imputation were carried out via SHAPEIT2 and IMPUTE4. We conducted association analyses with the same three dichotomous outcomes as in the GWAS on imputed dosage files.

Conditional analyses were conducted via the stepwise inclusion of variants with the lowest P -value as covariates in logistic regression models until no signal below $P < 1e^{-6}$ was left. To further account for the complex LD structure of the MHC we conducted additional conditional analysis by calculating the *C4A* and *C4B* expression levels for each individual based on imputed *C4* alleles. We have used the formula proposed by Sekar and colleagues (2016): $C4A$ expression = $(0.47 * AL) + (0.47 * AS) + (0.20 * BL)$; $C4B$ expression = $(1.03 * BL) + (0.88 * BS)$.”

6. In TWAS analysis (Supplementary Table 10), the authors reported many genes in the MHC region. If the authors believe that the HLA alleles and C4 copy number variations causally contribute to MG development, they need to be cautious about reporting TWAS results in the HLA region. The very long-range LD structure in the HLA region produces many falsely positive TWAS results.

In response to your feedback (and a similar one by another reviewer), we've completely excluded the extended MHC region (25–35Mb) from the TWAS. We agree that our initially presented TWAS results had identified MHC genes that could be in substantial linkage disequilibrium, and, as you correctly pointed out, we've addressed the MHC locus in separate analyses.

Results

Adapted paragraph:

"TWAS using the individual tissue-based prediction matrix of gene expression identified 138 significant genes whose transcript expression was significantly associated with MG. Permutation and co-localization tests identified 56 and 45 gene hits below the significance threshold. Combining TWAS, permutation, and co-localization tests, we identified 24 significant unique genes with high confidence. The TWAS highlighted individual genes in the highly complex loci on chromosome 12 and 17, as well as genes in additional loci that did not reach genome-wide significance in the GWAS, including *PAPPA* and *EPS15L1* (Table S10, Figure S12-14)."

Methods

Adapted paragraph:

"Transcriptome-wide association study (TWAS) was performed using FUSION (Gusev et al., 2016) to predict tissue-specific gene expression based on the MG discovery GWAS summary statistics excluding the extended MHC region (Chromosome 6: 25-35Mb). Gene expression weights from three Genotype-Tissue Expression (GTEx) version 8 (The GTEx Consortium, 2020) tissues were used for transcriptomic imputation and association testing: Muscle Skeletal ($n = 8,602$), Nerve Tibial ($n = 11,360$), and Whole Blood ($n = 8,059$). Gene associations were considered significant at a P -value $< 0.05/N$ of genes per tissue. A permutation test, which shuffles the quantitative trait loci (QTL) weights ($N_{\max} = 1000$) was performed to correct inflated associations from by-chance QTL co-localization. Co-localization was computed for genes with a P -value $< 1e^{-5}$. A posterior probability value ≥ 0.75 was considered as evidence for the expression-QTL-GWAS pair influencing both the expression and the GWAS trait in a particular region."

Minor comments:

1. Could the authors describe 12 cohorts in detail? Did the previous GWAS already analyze these cohorts? Or, any new cohorts or samples were added in this study?

We have included more detailed information on the individual sample size and previous publications (PMID IDs) of the 12 included cohorts in the supplementary tables. Details on sample ascertainment and institutional review board approval are included in the supplementary material. We have adapted the following paragraph:

Methods

Adapted paragraph:

"Information on previous publications, sample sizes and phenotypes per cohort are shown in Table S20-22."

2. How did the authors calculate the "effective N case", which appears multiple times on pages 7 and 8? There are several different ways to estimate these indices, so the authors need to clearly explain how they calculated it and provide a reference for it.

Thank you for pointing out that we did not report the formula for the effective sample size we report in the results. We have added the following paragraph to the methods section of the manuscript:

Methods

Added paragraph:

"We calculated half of the effective sample size (N_{eff}) for each cohort separately using the following formula: $(4 \times N_{\text{cases}} \times N_{\text{controls}} / (N_{\text{cases}} + N_{\text{controls}})) / 2$ and summed it across all cohorts. "

REVIEWERS' COMMENTS

Reviewer #1 (Remarks to the Author):

Comments addressed.

Reviewer #2 (Remarks to the Author):

I applaud the authors for carefully addressing all reviewers' comments. Also interesting to see comments that were similar across reviewers. I had one small question: when reporting SNP based h^2 on liability scale, the authors also need to report the life time risk for MG used to convert observed to liability in the methods section with one or two references. This can have quite some influence on results.

Reviewer #3 (Remarks to the Author):

The authors fairly addressed my concerns. I do not have further comments.